# GENERALISED LINEAR MODELS IN DEEP BAYESIAN RL WITH LEARNABLE BASIS FUNCTIONS

## ABSTRACT

Bayesian Reinforcement Learning (BRL) provides a framework for generalisation of Reinforcement Learning (RL) problems from its use of Bayesian task parameters in the transition and reward models. However, classical BRL methods assume known forms of transition and reward models, reducing their applicability in real-world problems. As a result, recent deep BRL methods have started to incorporate model learning, though the use of neural networks directly on the joint data and task parameters requires optimising the Evidence Lower Bound (ELBO). ELBOs are difficult to optimise and may result in indistinctive task parameters, hence compromised BRL policies. To this end, we introduce a novel deep BRL method, **G**eneralised **Li**near Models in Deep **B**ayesian **RL** with Learnable Basis Functions (**GLiBRL**), that enables efficient and accurate learning of transition and reward models, with fully tractable marginal likelihood and Bayesian inference on task parameters and model noises. On challenging MetaWorld ML10 and ML45 benchmarks, GLiBRL improves the success rate of one of the state-of-the-art deep BRL methods, VariBAD, by up to $2.7\times$. Comparing against representative or recent deep BRL / Meta-RL methods, such as MAML, $\text{RL}^2$, SDVT, TrMRL and ECET, GLiBRL also demonstrates its low-variance and decent performance consistently.

## 1 INTRODUCTION

Reinforcement Learning (RL) algorithms have great potentials to enable robots to act intelligently without supervisions from humans. A well-known issue with RL algorithms is they generalise poorly to unseen tasks. Most standard RL algorithms by their designs do not consider possible variations in the transition and reward models, hence fail to adapt to new tasks whose models might be different from that of training environments.

Bayesian Reinforcement Learning (BRL) is an effective framework that can be used to improve the generalisation of RL. Instead of ignoring possible variations in transition and reward models, BRL methods explicitly take them into considerations by assuming parametric distributions of models and performing Bayesian inference on the parameters (Ghavamzadeh et al., 2015). Different parameters indicate different transition and reward models, hence implicitly encode various tasks. To solve BRL problems, many previous works use planners (Poupart et al., 2006; Guez et al., 2013) that search for Bayes-optimal policies. These methods are often limited in their scalability. Moreover, they require full information about the forms of transition and reward models, which restricts generalisation across different tasks.

Hence, recent deep BRL methods (Rakelly et al., 2019; Zintgraf et al., 2021) enable model learning by optimising the marginal likelihood of the data. However, most of the deep BRL methods do not support tractable Bayesian inference on the task parameters, because of the direct use of neural networks on the joint data and parameters. As a result, the exact marginal likelihood of the data is also not tractable and cannot be optimised directly. To this end, deep BRL methods adopt variational inference to optimise the evidence lower bound (ELBO) instead. However, the optimisation of ELBO is not an easy task as it may face issues such as high-variance Monte Carlo estimates, amortisation gaps (Cremer et al., 2018) and posterior collapse (Bowman et al., 2016; Dai et al., 2020). Such issues can preclude BRL methods from obtaining meaningful and distinctive distributions of task parameters, which are crucial to smooth Bayesian learning.

To alleviate the above issues, we propose **G**eneralised **Li**near Models in Deep **B**ayesian **RL** with Learnable Basis Functions (**GLiBRL**). GLiBRL assumes a generalised linear relation between task parameters and features of the data, computed via learnable basis functions. Such modelling allows exact posterior inference and marginal likelihood to be computed, avoiding optimising the challenging ELBO objective. Furthermore, GLiBRL generalises previous works such as (Harrison et al., 2018b) in that GLiBRL also performs inference on the noise, reducing the error of predictions in unseen tasks.

The performance of GLiBRL is evaluated with the challenging MetaWorld (Yu et al., 2021; McLean et al., 2025) benchmark and compared with standard baselines and recent methods, including deep BRL methods, VariBAD (Zintgraf et al., 2021) and SDVT (Lee et al., 2023), and Meta-Reinforcement Learning (Beck et al., 2023b) methods, including RL$^2$ (Wang et al., 2016; Duan et al., 2016), MAML (Finn et al., 2017), TrMRL (Melo, 2022) and ECET (Shala et al., 2025). GLiBRL has improved the success rate of one of the state-of-the-art deep BRL methods, VariBAD, by up to $2.7\times$, and has consistently outperformed all listed methods and demonstrated low-variance behaviours in the most complex subset of MetaWorld, ML45.

## 2 BACKGROUND

### 2.1 MARKOV DECISION PROCESSES AND REINFORCEMENT LEARNING

Markov Decision Processes (MDPs) are 5-tuples defined as $\mathcal{M} = (\mathcal{S}, \mathcal{A}, R, T, \gamma)$, where $\mathcal{S}$ is the set of *states*; $\mathcal{A}$, the set of *actions*; $R(s_t, a_t, s_{t+1}, r_{t+1}) = p(r_{t+1}|s_t, a_t, s_{t+1})$, the *reward* function; $T(s_t, a_t, s_{t+1}) = p(s_{t+1}|s_t, a_t)$, the *transition* function; and $\gamma \in (0, 1]$ a discount factor. In the above definition, $s_t \in \mathcal{S}, s_{t+1} \in \mathcal{S}, a_t \in \mathcal{A}$ and $r_{t+1} \in \mathbb{R}$. The goal of solving an MDP is to find a policy $\pi(a_t|s_t) : \mathcal{S} \to \mathcal{A}$, such that the expected return for a finite horizon $H > 0$, $\mathcal{J}(\pi) = \mathbb{E}_{T,\pi,R}[\sum_{t=0}^{H} \gamma^t R(s_t, a_t, s_{t+1}, r_{t+1})]$, is maximised.

While MDPs assume fully known transition and reward functions $T$ and $R$, standard Reinforcement Learning (RL) problems optimise the same objective, however, with at least one of the $T$ and $R$ unknown but can be learnt from data samples.

### 2.2 BAYES-ADAPTIVE MDPs AND BAYESIAN REINFORCEMENT LEARNING

Bayes-Adaptive MDPs (BAMDPs) (Duff, 2002; Ghavamzadeh et al., 2015) is a Bayesian framework for solving RL. Compared to standard MDPs, BAMDPs assume *known* forms of functions of $T$ and / or $R$, but parameterised by *unknown* parameters $\theta_T \in \Theta_T$ and / or $\theta_R \in \Theta_R$.

In BAMDPs, distributions (or, *beliefs*) $b_t = p(\theta_{T,t}, \theta_{R,t}) \in \mathcal{B}_T \times \mathcal{B}_R$ are placed on the unknown parameters, and updated to posteriors $b_{t+1} = p(\theta_{T,t+1}, \theta_{R,t+1})$ with Bayesian inference.

To efficiently use existing MDP frameworks, the beliefs can be absorbed into the original state space to form hyper-states $\mathcal{S}^+ = \mathcal{S} \times \mathcal{B}_T \times \mathcal{B}_R$. Hence, BAMDPs can be defined as 5-tuple $(\mathcal{S}^+, \mathcal{A}, R^+, T^+, \gamma)$ MDPs, where

$$
\begin{aligned}
T^+(s_t^+, a_t, s_{t+1}^+) &= p(s_{t+1}, b_{t+1}|s_t, b_t, a_t) \\
&= \mathbb{E}_{\theta_{T,t} \sim b_t}\left[p(s_{t+1}|s_t, a_t, \theta_{T,t})\right] \cdot \delta(b_{t+1} = p(\theta_{T,t+1}, \theta_{R,t+1}))
\end{aligned}
\tag{1}
$$

$$
\begin{aligned}
R^+(s_t^+, a_t, s_{t+1}^+, r_{t+1}) &= p(r_{t+1}|s_t, b_t, a_t, s_{t+1}, b_{t+1}) \\
&= \mathbb{E}_{\theta_{R,t+1} \sim b_{t+1}}\left[p(r_{t+1}|s_t, a_t, s_{t+1}, \theta_{R,t+1})\right]
\end{aligned}
\tag{2}
$$

The hyper-transition function (Equation 1) consists of the $\theta_T$-parameterised expected regular MDP transition and a deterministic posterior update specified by the Dirac delta function $\delta(b_{t+1} = p(\theta_{T,t+1}, \theta_{R,t+1}))$. The hyper-reward function consists of the $\theta_R$-parameterised expected regular MDP reward function. Accordingly, the expected return to maximise becomes $J^+(\pi^+) = \mathbb{E}_{T^+, \pi^+, R^+}[\sum_{t=0}^{H^+} \gamma^t R^+(s_t^+, a_t, s_{t+1}^+, r_{t+1})]$, where $H^+ > 0$ is the BAMDP horizon, and $\pi^+ : \mathcal{S}^+ \to \mathcal{A}$ is the policy of BAMDPs. Traditionally, problems that require solving BAMDPs are named Bayesian Reinforcement Learning (BRL). Aside from their generalisability, BRL methods are also recognised for offering principled approaches to the exploration–exploitation problem in RL (Ghavamzadeh et al., 2015).

However, classical BRL methods (Poupart et al., 2006; Guez et al., 2013; Tziortziotis et al., 2013) assume that the forms of transition $T^+$ and reward $R^+$ models are fully known, despite being parameterised by unknown parameters. These methods are not sufficiently flexible in scenarios where the forms of $T^+$ and $R^+$ are not known a priori. A rough guess of the forms, however, may lead to significant underfit (e.g., assuming linear transitions while the ground truth is quadratic).

To generalise classical BRL methods, Hidden-Parameter MDPs (HiP-MDPs) (Doshi-Velez & Konidaris, 2016; Killian et al., 2017; Yao et al., 2018) have started to learn the forms of models through performing Bayesian inference on the weights. Doshi-Velez & Konidaris (2016) proposed HiP-MDPs with Gaussian Processes (GPs) to learn the basis functions for approximating transition models. Afterwards, Killian et al. (2017) discovered the poor scalability from the use of GPs, and applied Bayesian Neural Networks (BNNs) in HiP-MDPs for larger scale problems. (Yao et al., 2018) proposed to fix the weights of BNNs during evaluation for improved efficiency, at the cost of losing most of the test-time Bayesian features. These works focus on performing Bayesian inference on the weights of BNNs, which does not scale well with the size of BNN and is empirically demonstrated by (Yang et al., 2019). Reward functions in the HiP-MDP setting are also assumed to be known, which is generally infeasible in real-world applications .

On the other hand, recent deep BRL methods (Harrison et al., 2018a; Rakelly et al., 2019; Zintgraf et al., 2021) adopt scalable regular deep neural networks, as Bayesian features remain by performing (approximate) Bayesian inference on task parameters $\theta_T, \theta_R$ directly. GLiBRL follows this line of works for scalability and also the more general assumption of unknown reward functions. We briefly introduce how the learning of the forms of transition and reward models is done in the deep BRL setting. The deep BRL agent is provided with MDPs with unknown transitions $T$ and / or rewards $R$, and simulators from which the agent can obtain samples of tuples, known as *contexts* (Rakelly et al., 2019; Zintgraf et al., 2021). The context at step $t$ is defined as $c_t := \left( s_t \in \mathbb{R}^{1 \times D_S}, a_t \in \mathbb{R}^{1 \times D_A}, s_{t+1} \in \mathbb{R}^{1 \times D_S}, r_{t+1} \in \mathbb{R}^{1 \times 1} \right)$ . Denote those MDPs as $\mathcal{M}$, contexts obtained from $\mathcal{M}$ as $\mathcal{C}^{\mathcal{M}}$ where

$$\mathcal{C}^{\mathcal{M}} := \{c_t\}_{t=1}^N = \left\{ s_t \in \mathbb{R}^{1 \times D_S}, a_t \in \mathbb{R}^{1 \times D_A}, s_{t+1} \in \mathbb{R}^{1 \times D_S}, r_{t+1} \in \mathbb{R}^{1 \times 1} \right\}_{t=1}^N$$

During deep BRL training, assume we have access to MDPs $\{\mathcal{M}_i\}_{i=1}^M$ and the joint contexts $\mathcal{C} = \bigcup_{i=1}^M \mathcal{C}^{\mathcal{M}_i}$. The objective of most recent deep BRL methods (Rakelly et al., 2019; Perez et al., 2020; Zintgraf et al., 2021) is to maximise the marginal log-likelihood of the joint context[1]

$$\log p_{\zeta, \phi_T, \phi_R}(\mathcal{C}) = \sum_{i=1}^M \log p_{\zeta, \phi_T, \phi_R}(\mathcal{C}^{\mathcal{M}_i})$$
$$= \sum_{i=1}^M \log \iint p_\zeta(\theta_T, \theta_R) p_{\phi_T, \phi_R}(\mathcal{C}^{\mathcal{M}_i} | \theta_T, \theta_R) d\theta_T d\theta_R \tag{3}$$

where $\zeta$ is a neural network to learn the parameters of the prior distribution $p(\theta_T, \theta_R)$, $p_{\phi_T, \phi_R}(\mathcal{C}^{\mathcal{M}_i} | \theta_T, \theta_R) \propto \prod_{t=1}^N p_{\phi_T}(s_{t+1} | s_t, a_t, \theta_T) p_{\phi_R}(r_{t+1} | s_t, a_t, s_{t+1}, \theta_R)$, and $\phi_T, \phi_R$ are neural networks to learn the forms of transition and reward functions.

For the ease of learning, $p_{\phi_T}$ and $p_{\phi_R}$ are generally assumed to be Gaussian with mean and diagonal covariance determined by the output of neural networks $\phi_T, \phi_R$, and the prior $p_\zeta(\theta_T, \theta_R)$ is also assumed to be a Gaussian. However, note that even with these simplifications, Equation 3 is still not tractable as $\theta_T, \theta_R$ are not linear with respect to contexts, because of the use of neural networks. Fortunately, variational inference provides a lower bound to Equation 3, named Evidence Lower Bound (ELBO) that can be used as an approximate objective function (proof see Appendix A.1):

$$\log p_{\zeta, \phi_T, \phi_R}(\mathcal{C}) \geq \sum_{i=1}^M \mathbb{E}_q \left[ \log p_{\phi_T, \phi_R}(\mathcal{C}^{\mathcal{M}_i} | \theta_T, \theta_R) \right] - D_{KL} \left( q(\theta_T, \theta_R | \mathcal{C}^{\mathcal{M}_i}) \| p_\zeta(\theta_T, \theta_R) \right) \tag{4}$$

where $D_{KL}(\cdot || \cdot)$ is the KL-divergence, and $q(\cdot)$ is an approximate Gaussian posterior of $\theta_T, \theta_R$. An optimised $\log p_{\zeta, \phi_T, \phi_R}(\mathcal{C})$ will bring in models for transitions and rewards, with which both model-free (Rakelly et al., 2019; Zintgraf et al., 2021) and model-based (Guez et al., 2013; Harrison et al., 2018a) methods can be applied for learning the BRL policy.

---

[1]Henceforth, we drop the dependence on time step $t$ of $\theta_T, \theta_R$ for brevity.

ELBO-like objectives enable the learning of transition and reward models. However, ELBOs are challenging to optimise, for known issues such as high-variance Monte Carlo estimates, amortisation gaps (Cremer et al., 2018) and posterior collapse (Bowman et al., 2016; Dai et al., 2020). Unoptimised ELBOs may result in scenarios where learnt latent representations (e.g., $\theta_T, \theta_R$ in BRL) are not meaningful and distinctive. Different from other tasks where meaningful latent representations are less important, BRL policies determine the next action to perform heavily dependent on continually updated distributions of latent representations. Indistinctive latent representations, hence meaningless posterior updates will substantially harm the performance of BRL policies.

Aside from issues in ELBO, it is also concerning how previous methods compute the posterior $q(\theta_T, \theta_R | \mathcal{C}^{\mathcal{M}_i})$. As $\mathcal{C}^{\mathcal{M}_i}$ contains variable and large number of contexts, it is inefficient to directly use it as a conditional variable. Instead, Rakelly et al. (2019) applied factored approximation so that $q(\theta_T, \theta_R | \mathcal{C}^{\mathcal{M}_i}) \approx \prod_{t=1}^{N} \mathcal{N}(\theta_T, \theta_R | g([\mathcal{C}^{\mathcal{M}_i}]_t))$, where $g(\cdot)$ is a neural network that takes the $t$-th context in $\mathcal{C}^{\mathcal{M}_i}$ as the input and returns the mean and covariance of a Gaussian as the output. From Bayes rule, $q(\theta_T, \theta_R | \mathcal{C}^{\mathcal{M}_i}) = p(\theta_T, \theta_R)^{1-N} \prod_{t=1}^{N} q(\theta_T, \theta_R | [\mathcal{C}^{\mathcal{M}_i}]_t)$. This is to say, for the approximation to be accurate, $\mathcal{N}(\theta_T, \theta_R | g([\mathcal{C}^{\mathcal{M}_i}]_t)) = p(\theta_T, \theta_R)^{1/N-1} q(\theta_T, \theta_R | [\mathcal{C}^{\mathcal{M}_i}]_t)$. From the left-hand-side, $g(\cdot)$ tries to predict the mean and the covariance regardless of the prior, while the right-hand-side has the prior involved, meaning the same $\mathcal{N}(\theta_T, \theta_R | g([\mathcal{C}^{\mathcal{M}_i}]_t))$ gets implicitly assigned with different targets as $N$ increases. This may result in inaccuracies of the approximation and unstable training. On the other hand, Zintgraf et al. (2021) summarise $\mathcal{C}^{\mathcal{M}_i}$ with RNNs to get hidden variables $h$, and compute $q(\theta_T, \theta_R | h)$. Despite its simplicity, it has been shown in Rakelly et al. (2019) that permutation-variant structures like RNNs may lead to worse performance.

# 3 GENERALISED LINEAR MODELS IN BAYESIAN RL WITH LEARNABLE BASIS FUNCTIONS

Previous deep BRL methods perform inaccurately approximated posterior updates and optimise challenging ELBOs. Both issues may lead to incorrect distributions of task parameters, compromising the performance of BRL policies. To this end, we introduce our method, GLiBRL. GLiBRL features generalised linear models that enable fully tractable and permutation-invariant posterior update, hence closed-form marginal log-likelihood, without the need to evaluate and optimise the ELBO. The linear assumption seems strong, but basis functions still enable linear models to learn non-linear transitions and rewards. The basis functions that maps from the raw data $\mathcal{C}^{\mathcal{M}_i}$ to the feature space are made *learnable* from the marginal log-likelihood, instead of being chosen arbitrarily, allowing for efficient learning under low-dimensional feature space. We elaborate on the learning of the forms of transition and reward functions in Section 3.1 and discuss efficient online policy learning in Section 3.2. The full GLiBRL algorithm is demonstrated in Algorithm 1.

## 3.1 LEARNING THE FORMS OF TRANSITIONS AND REWARDS

We rewrite $\mathcal{C}^{\mathcal{M}_i} = (\mathbf{S}_i \in \mathbb{R}^{N \times D_S}, \mathbf{A}_i \in \mathbb{R}^{N \times D_A}, \mathbf{S}'_i \in \mathbb{R}^{N \times D_S}, \mathbf{r}_i \in \mathbb{R}^{N \times 1})$ for compactness, where $N$ is the number of context and $D_S, D_A$ are the dimensions of the state and action space. We further let $\theta_T = (T_\mu \in \mathbb{R}^{D_T \times D_S}, T_\sigma \in \mathbb{R}^{D_S \times D_S})$, $\theta_R = (R_\mu \in \mathbb{R}^{D_R \times 1}, R_\sigma \in \mathbb{R}^{1 \times 1})$, where $D_T, D_R$ are task dimensions. Note we explicitly perform Bayesian inference on $T_\sigma, R_\sigma$, instead of assuming known model noises. We make the following approximation:

$$p_{\phi_T}(\mathbf{S}'_i | \mathbf{S}_i, \mathbf{A}_i, \theta_T) = \mathcal{MN}\left(\mathbf{S}'_i | \mathbf{C}_T T_\mu, \mathbf{I}_N, T_\sigma\right) \tag{5}$$

$$p_{\phi_R}(\mathbf{r}_i | \mathbf{S}_i, \mathbf{A}_i, \mathbf{S}'_i, \theta_R) = \mathcal{MN}\left(\mathbf{r}_i | \mathbf{C}_R R_\mu, \mathbf{I}_N, R_\sigma\right) \tag{6}$$

where $\mathbf{C}_T, \mathbf{C}_R$ [2] are features of contexts $\mathbf{S}_i, \mathbf{A}_i, \mathbf{S}'_i$ computed through neural networks which act as learnable basis functions

$$\mathbf{C}_T \in \mathbb{R}^{N \times D_T} = \phi_T(\mathbf{S}_i, \mathbf{A}_i) \qquad \mathbf{C}_R \in \mathbb{R}^{N \times D_R} = \phi_R(\mathbf{S}_i, \mathbf{A}_i, \mathbf{S}'_i) \tag{7}$$

and $\mathcal{MN}(\mathbf{W} | \mathbf{X}, \mathbf{Y}, \mathbf{Z})$ defines a matrix normal distribution with random matrix $\mathbf{W}$, mean $\mathbf{X}$, row covariance $\mathbf{Y}$ and column covariance $\mathbf{Z}$. Different from other deep BRL methods such as PEARL (Rakelly et al., 2019) and VariBAD (Zintgraf et al., 2021), GLiBRL does not place neural networks on the joint contexts and task parameters (e.g., $\phi_T(\mathbf{S}_i, \mathbf{A}_i, \theta_T)$), in order for tractable inference.

---

[2]Dependence on $i$ of $\mathbf{C}_T, \mathbf{C}_R$ is omitted for clarity. This also applies to $\mathbf{M}'_T, \mathbf{\Xi}'_T, \mathbf{\Omega}'_T, \mathbf{M}'_R, \mathbf{\Xi}'_R, \mathbf{\Omega}'_R$.

---

**Algorithm 1:** GLiBRL

---

Initialise: policy $\pi_\psi^+$, horizon $H$, $\phi_T, \phi_R$;

**while** *Training* **do**

    Sample $K$ MDPs $\{\mathcal{M}_i\}_{i=1}^K$;

    Initialise: contexts $\mathcal{C} = \{\}$;

    // collecting contexts

    **for** $i \in \{1, 2, \cdots, K\}$ **do**

        Initialise: contexts in $\mathcal{M}_i$, $\mathcal{C}^{\mathcal{M}_i} = \{\}$, state $s$;

        **for** $t < H$ **do**

            $b \leftarrow p_{\phi_T, \phi_R}(\theta_T, \theta_R | \mathcal{C}^{\mathcal{M}_i})$;

            $a \sim \pi_\psi^+(a|s, b)$;

            Execute $a$ from $s$ in $\mathcal{M}_i$ to get $s', r$;

            $s \leftarrow s'$;

            $\mathcal{C}^{\mathcal{M}_i} \leftarrow \mathcal{C}^{\mathcal{M}_i} \cup \{s, a, s', r\}$;

        $\mathcal{C} = \mathcal{C} \cup \mathcal{C}^{\mathcal{M}_i}$

    // learning policy, transition and reward models

    **while** *Learning* **do**

        Sample $\mathcal{D} \subseteq \mathcal{C}$;

        $\psi \leftarrow \psi - \nabla_\psi \mathcal{L}_{\text{Policy}}(\mathcal{D})$;

        $\phi_T \leftarrow \phi_T - \nabla_{\phi_T} \mathcal{L}_{\text{model}}(\mathcal{D})$;

        $\phi_R \leftarrow \phi_R - \nabla_{\phi_R} \mathcal{L}_{\text{model}}(\mathcal{D})$;

---

Assuming the independence of $\theta_T$ and $\theta_R$, dropping the neural network $\zeta$ of the prior, Equation 3 can be written as

$$
\log p_{\phi_T, \phi_R}(\mathcal{C}) = \sum_{i=1}^M \log \int p(\theta_T) p_{\phi_T}(\mathbf{S}_i' | \mathbf{S}_i, \mathbf{A}_i, \theta_T) d\theta_T
$$

$$
+ \sum_{i=1}^M \log \int p(\theta_R) p_{\phi_R}(\mathbf{r}_i | \mathbf{S}_i, \mathbf{A}_i, \mathbf{S}_i', \theta_R) d\theta_R + \text{const.}
$$

(8)

Because of the linear relationship between $\theta_T, \theta_R$ and features of the contexts, we can place Normal-Wishart priors conjugate to matrix normals on $\theta_T, \theta_R$ for tractable inference

$$
p(\theta_T) = p(T_\mu, T_\sigma) = \mathcal{MN}(T_\mu | \mathbf{M}_T \in \mathbb{R}^{D_T \times D_S}, \mathbf{\Xi}_T^{-1} \in \mathbb{R}^{D_T \times D_T}, T_\sigma)
$$
$$
\cdot \mathcal{W}(T_\sigma^{-1} | \mathbf{\Omega}_T^{-1} \in \mathbb{R}^{D_S \times D_S}, \nu_T \in \mathbb{R}_{++})
$$

(9)

$$
p(\theta_R) = p(R_\mu, R_\sigma) = \mathcal{MN}(R_\mu | \mathbf{M}_R \in \mathbb{R}^{D_R \times 1}, \mathbf{\Xi}_R^{-1} \in \mathbb{R}^{D_R \times D_R}, R_\sigma)
$$
$$
\cdot \mathcal{W}(R_\sigma^{-1} | \mathbf{\Omega}_R^{-1} \in \mathbb{R}^{1 \times 1}, \nu_R \in \mathbb{R}_{++})
$$

(10)

where $\mathcal{W}(\mathbf{W}|\mathbf{X}, \nu)$ defines a Wishart distribution on positive definite random matrix $\mathbf{W}$ with scale $\mathbf{X}$ and degrees of freedom $\nu$. It has been shown in Appendix A.2 that the posteriors are also Normal-Wishart distributions

$$
p_{\phi_T}(\theta_T | \mathbf{S}_i, \mathbf{A}_i, \mathbf{S}_i') = \mathcal{MN}(T_\mu | \mathbf{M}_T', \mathbf{\Xi}_T'^{-1}, T_\sigma) \cdot \mathcal{W}(T_\sigma^{-1} | \mathbf{\Omega}_T'^{-1}, \nu_T')
$$

(11)

$$
p_{\phi_R}(\theta_R | \mathbf{S}_i, \mathbf{A}_i, \mathbf{S}_i', \mathbf{r}) = \mathcal{MN}(R_\mu | \mathbf{M}_R', \mathbf{\Xi}_R'^{-1}, R_\sigma) \cdot \mathcal{W}(R_\sigma^{-1} | \mathbf{\Omega}_R'^{-1}, \nu_R')
$$

(12)

where

$$
\begin{aligned}
\mathbf{M}_T' &= \mathbf{\Xi}_T'^{-1} \left[ \mathbf{C}_T^{\mathrm{T}} \mathbf{S}' + \mathbf{\Xi}_T \mathbf{M}_T \right] & \mathbf{M}_R' &= \mathbf{\Xi}_R'^{-1} \left[ \mathbf{C}_R^{\mathrm{T}} \mathbf{r} + \mathbf{\Xi}_R \mathbf{M}_R \right] \\
\mathbf{\Xi}_T' &= \mathbf{C}_T^{\mathrm{T}} \mathbf{C}_T + \mathbf{\Xi}_T & \mathbf{\Xi}_R' &= \mathbf{C}_R^{\mathrm{T}} \mathbf{C}_R + \mathbf{\Xi}_R \\
\mathbf{\Omega}_T' &= \mathbf{\Omega}_T + \mathbf{S}'^{\mathrm{T}} \mathbf{S}' & \mathbf{\Omega}_R' &= \mathbf{\Omega}_R + \mathbf{r}^{\mathrm{T}} \mathbf{r} \\
&\quad + \mathbf{M}_T^{\mathrm{T}} \mathbf{\Xi}_T \mathbf{M}_T - \mathbf{M}_T'^{\mathrm{T}} \mathbf{\Xi}_T' \mathbf{M}_T' & &\quad + \mathbf{M}_R^{\mathrm{T}} \mathbf{\Xi}_R \mathbf{M}_R - \mathbf{M}_R'^{\mathrm{T}} \mathbf{\Xi}_R' \mathbf{M}_R' \\
\nu_T' &= \nu_T + N & \nu_R' &= \nu_R + N
\end{aligned}
$$

(13)

Thus, we can find a closed-form marginal log-likelihood (proof in Appendix A.3):

$$\log p_{\phi_T, \phi_R}(\mathcal{C}) = -\frac{1}{2}\sum_{i=1}^{M} D_S \log|\mathbf{\Xi}'_T| + \nu'_T \log|\frac{1}{2}\mathbf{\Omega}'_T| + \log|\mathbf{\Xi}'_R| + \nu'_R \log|\frac{1}{2}\mathbf{\Omega}'_R| + \text{const.} \quad (14)$$

Equation 14 is to be maximised with related to $\mathbf{C}_T$ and $\mathbf{C}_R$, hence $\phi_T$ and $\phi_R$. We add squared Frobenius norms $\|\mathbf{C}_T\|_F^2$ and $\|\mathbf{C}_R\|_F^2$ to Equation 14 as regularisations, the effect of which being discussed in Appendix A.8. The regularised loss function is defined as

$$\mathcal{L}_{\text{model}} := -\log p_{\phi_T, \phi_R}(\mathcal{C}) + \lambda_T\|\mathbf{C}_T\|_F^2 + \lambda_R\|\mathbf{C}_R\|_F^2 \quad (15)$$

where $\lambda_T > 0$ and $\lambda_R > 0$ are hyperparameters. We note that $\mathcal{L}_{\text{model}}$ can be directly minimised with gradient descent, without the need to evaluate and optimise the ELBO.

## 3.2 Learning the Policy

A natural follow-up question is how to collect the contexts $\mathcal{C}$. We adopt a learnable BAMDP policy $\pi_\psi^+(a_t|s_t, b_t)$ parameterised by the neural network $\psi$. $\mathcal{C}$ can be collected by rolling out $\pi_\psi^+$ in training MDPs $\{\mathcal{M}_i\}_{i=1}^{M}$, and $\psi$ can be updated using $\mathcal{C}$ with model-free or model-based RL methods.

$\pi_\psi^+(a_t|s_t, b_t)$ takes beliefs $b_t$ as its input. To roll out $\pi_\psi^+$ online, the prior $b_t$ needs to be continually updated to the posterior $b_{t+1}$ from the new context $c = \{s_t, a_t, s'_t, r_{t+1}\}$, using the learnt transition and reward models. Therefore, fast posterior update is crucial for efficient context collections. One of the most time-consuming part in Equation 13 is the inversion of $\mathbf{\Xi}'_T$ and $\mathbf{\Xi}'_R$, which is of time complexity $O(D_T^3)$ and $O(D_R^3)$, respectively. Fortunately, with the matrix inversion lemma

$$\mathbf{\Xi}'^{-1}_T = \left(\mathbf{C}_T^{\mathsf{T}}\mathbf{C}_T + \mathbf{\Xi}_T\right)^{-1} = \mathbf{\Xi}_T^{-1} - \mathbf{\Xi}_T^{-1}\mathbf{C}_T^{\mathsf{T}}\left(\mathbf{I}_N + \mathbf{C}_T\mathbf{\Xi}_T^{-1}\mathbf{C}_T^{\mathsf{T}}\right)^{-1}\mathbf{C}_T \quad (16)$$

$$\mathbf{\Xi}'^{-1}_R = \left(\mathbf{C}_R^{\mathsf{T}}\mathbf{C}_R + \mathbf{\Xi}_R\right)^{-1} = \mathbf{\Xi}_R^{-1} - \mathbf{\Xi}_R^{-1}\mathbf{C}_R^{\mathsf{T}}\left(\mathbf{I}_N + \mathbf{C}_R\mathbf{\Xi}_R^{-1}\mathbf{C}_R^{\mathsf{T}}\right)^{-1}\mathbf{C}_R \quad (17)$$

When updating the belief online with the new context, $\mathbf{C}_T \in \mathbb{R}^{1 \times D_T}$ and $\mathbf{C}_R \in \mathbb{R}^{1 \times D_R}$. Hence, $\left(\mathbf{I}_N + \mathbf{C}_T\mathbf{\Xi}_T^{-1}\mathbf{C}_T^{\mathsf{T}}\right)^{-1}$ and $\left(\mathbf{I}_N + \mathbf{C}_R\mathbf{\Xi}_R^{-1}\mathbf{C}_R^{\mathsf{T}}\right)^{-1}$ are reduced to reciprocals of scalars. Keeping track of the inverse of priors and posteriors, the inversion only takes $O(D_T^2)$ and $O(D_R^2)$. The full online update of all parameters takes $O(\max\{D_T^2 D_S, D_S^2 D_T\})$ and $O(D_R^2)$.

## 4 Related Work

**ALPaCA**. First, we discuss the most relevant work, ALPaCA (Harrison et al., 2018b). ALPaCA is an efficient and flexible online Bayesian linear regression framework, which also involves Bayesian linear models with learnable basis functions. ALPaCA initially was not proposed as an BRL method, though follow-up work such as CAMeLiD (Harrison et al., 2018a) uses controllers to compute the policy assuming known reward functions. Our method, GLiBRL, generalises ALPaCA and CAMeLiD in (1) ALPaCA and CAMeLiD assume a *known* noise in the likelihood function, instead of performing Bayesian inference, (2) ALPaCA and CAMeLiD are not evaluated in online BRL settings. They only investigated scenarios where offline contexts are available with unknown transitions and relatively simple known rewards. In Section 5, we will argue empirically that the assumption of known noises incurs error in both predictions of transitions and rewards.

**Reinforcement Learning**. RL methods can be categorised as model-free and model-based. We use the former in this paper to learn BRL policies as a large proportion of RL work is model-free, such as Trust-Region Policy Optimisation (TRPO) (Schulman et al., 2015), Proximal Policy Optimisation (PPO) (Schulman et al., 2017) and Soft Actor-Critic (SAC) (Haarnoja et al., 2018). As GLiBRL learns the models, model-based methods can also be used for improved sample efficiency.

**Hidden-Parameter MDPs**. Hidden-Parameter MDP (HiP-MDP), proposed by Doshi-Velez & Konidaris (2016), is a framework for parametric Bayesian Reinforcement Learning. HiP-MDP was initially modelled using Gaussian Processes (GPs). Killian et al. (2017) improved the scalability by replacing GPs with Bayesian Neural Networks (BNNs). The weights of BNNs are updated with new data during evaluation, which has been empirically shown inefficient (Yang et al., 2019). Yao

et al. (2018) mitigated the inefficiency by fixing the test-time weights of BNNs and optimising task parameters. Despite the improved speed, we have observed that this would divert the agent from following Bayes-optimal policies. The shared objectives in (Killian et al., 2017) and (Yao et al., 2018) correspond to approximate Bayesian inference on BNN weights, but not on task parameters. Optimising the objective on task parameters with fixed BNN weights is equivalently performing Maximum Likelihood Estimations (MLEs) on task parameters, immediately removing the Bayesian features (which is also mentioned in (Zintgraf et al., 2021)). Most recent parametric deep BRL (Rakelly et al., 2019; Zintgraf et al., 2021; Lee et al., 2023), including GLiBRL, are considered orthogonal to this line of works as they perform (approximate) Bayesian inference on task parameters directly, rather than on the weights of the neural networks. Furthermore, HiP-MDPs also assume known reward functions, while GLiBRL and methods such as (Rakelly et al., 2019; Zintgraf et al., 2021; Lee et al., 2023) do not have this limiting assumption.

**Classical Bayesian Reinforcement Learning**. As mentioned in Section 2, classical BRL methods assume known forms of transitions and rewards. Poupart et al. (2006) presented a Partially Observable MDP (POMDP) formulation of BRL and a sampling-based offline solver. Guez et al. (2013) proposed an online tree-based solver, applying posterior sampling (Strens, 2000; Osband et al., 2013) for efficiency. Both methods use solver for the (approximately) optimal policy using planners, which is orthogonal to GLiBRL. GLiBRL shares the idea of using generalised linear models with Tziortziotis et al. (2013), while differs in that Tziortziotis et al. (2013) *chooses* the basis function, instead of learning them. Even a simple non-linear basis function, such as quadratic function, may result in $O(d^2)$ dimensional feature space, where $d$ is the dimension of the raw input[3]. As demonstrated in Section 3, performing online Bayesian inference at least requires quadratic complexity with related to the feature dimension, meaning prohibitive $O(d^4)$ complexity is required for just a single inference step. In contrast, with learnt basis functions in GLiBRL, low-dimensional feature space are usually sufficient to capture the non-linearity, providing sufficient scalability.

**Meta-Reinforcement Learning**. Meta-Reinforcement Learning (Meta-RL) aims to learn policies from *seen* tasks that are capable of adapting to *unseen* tasks following similar task distributions (Beck et al., 2023b). According to Beck et al. (2023b), Meta-RL methods can be categorised as **(1) parameterised policy gradient (PPG)** methods (e.g., Finn et al. (2017); Yoon et al. (2018); Finn et al. (2018)) that learn by performing meta-policy gradients on the meta-parameterised policy; **(2) black-box methods** (e.g., Duan et al. (2016); Wang et al. (2016); Melo (2022); Shala et al. (2025)) that learn from summaries of histories and **(3) task-inference methods** (e.g., Rakelly et al. (2019); Zintgraf et al. (2021); Lee et al. (2023)) that learn from the belief of the task parameters (i.e., $\theta_T, \theta_R$ in GLiBRL). Most of the deep BRL methods can be viewed as task inference methods, hence a subset of Meta-RL. BRL differs from other Meta-RL methods in that it performs inference on task parameters, hence enjoying nice properties such as uncertainty quantification. As a result, BRL methods have natural integrations with model-based algorithms, hence significant to research fields such as control and planning under uncertainty.

## 5 EXPERIMENTS

In this section, we investigate the performance of GLiBRL using MetaWorld (Yu et al., 2021; McLean et al., 2025), one of the most famous and challenging meta-RL benchmark. The most recent and standardised version, MetaWorld-V3 (McLean et al., 2025), is used for fairness of comparisons. We focus on the most challenging subset of the MetaWorld benchmark, Meta-Learning 10 (ML10) and Meta-Learning 45 (ML45). ML10 / ML45 consists of 10 / 45 training tasks and 5 testing tasks. The testing tasks remain unseen during training. In all tasks, $D_S = 39$ and $D_A = 4$.

We follow the majority of the experiment settings in McLean et al. (2025). ML10 and ML45 experiments are run for 2e7 steps and 9e7 steps, respectively. We do not allow any adaptive steps during test time, hence evaluating the zero-shot performance. The zero-shot performance is critical to Bayesian RL and Meta-RL, as the goal of Meta-RL is to adapt with as little data as possible (Beck et al., 2023b). We compare the Inter-Quartile Mean (IQM) of the results, as suggested by Agarwal et al. (2021). Each experiment is run with a single A100 GPU[4] for 10 times.

---

[3]In GLiBRL, $d = D_S + D_A$ for transition models and $d = 2 \cdot D_S + D_A$ for reward models.

[4]A100 is not mandatory. GLiBRL is runnable with $\leq$ 8GB GPU memory, see Appendix A.10.

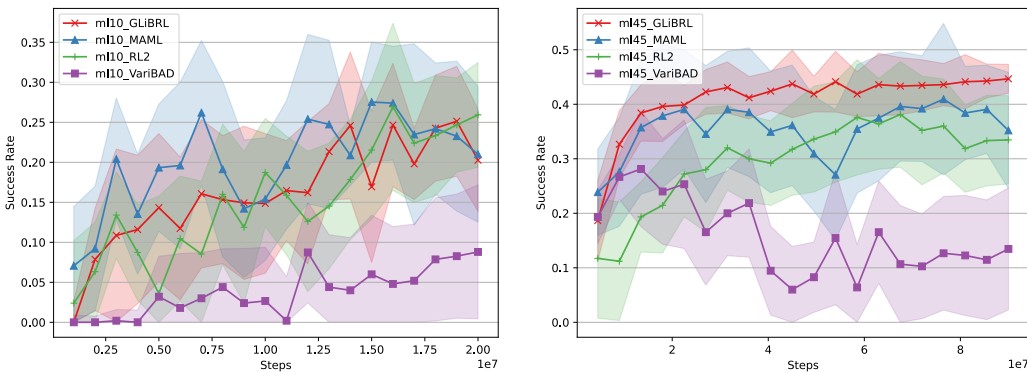

Figure 1: IQM and $95\%$ CI of testing success rate of GLiBRL, VariBAD, MAML and RL$^2$, with related to the number of training steps. Left: the ML10 benchmark; Right: the ML45 benchmark.

We list the comparators as follows. First, we compare GLiBRL to standard deep BRL and Meta-RL baselines, including deep BRL method VariBAD (Zintgraf et al., 2021), PPG-based Meta-RL methods MAML, and black-box Meta-RL method RL$^2$. Following McLean et al. (2025); Beck et al. (2023a), MAML learns its policy with TRPO, and VariBAD, RL$^2$ and GLiBRL use PPO. Our PPO implementation uses standard linear feature baseline (Duan et al., 2016), as suggested by McLean et al. (2025). Afterwards, we compare GLiBRL to recent deep BRL and Meta-RL baselines, including deep BRL method SDVT (Lee et al., 2023) and two Transformer-based (Vaswani et al., 2017) black-box Meta-RL methods, TrMRL (Melo, 2022) and ECET (Shala et al., 2025). Finally, we perform ablation studies on whether it is useful to place a Wishart distribution on model noises (i.e., comparing with ALPaCA). It is also worth mentioning why we do not compare with PEARL – it has been demonstrated empirically in Yu et al. (2021) that PEARL performs much worse than other methods. [5]

## 5.1 IMPLEMENTATION DETAILS

We report the details of our implementations of GLiBRL and other methods. For GLiBRL, we set task dimensions $D_T = 16$ and $D_R = 256$ in both ML10 / ML45. We demonstrate an analysis on the sensitivity of GLiBRL with related to task dimensions $D_T$ and $D_R$ in Appendix A.9. The model networks $\phi_T, \phi_R$ are Multi-Layer Perceptrons (MLPs) consisting of *feature* and *mixture* networks. Feature networks convert raw states and actions to features and are shared in $\phi_T$ and $\phi_R$. Mixture networks mix the state and action features, further improving the representativeness. The training of $\pi_\psi^+(a|s,b)$ requires representing the beliefs $b$ as parameters. Empirically, we find representing $b$ using flattened mean matrices $\mathbf{M}_T, \mathbf{M}_R$ have the best performance. As flattening $\mathbf{M}_T \in \mathbb{R}^{D_T \times D_S}$ directly results in a large number of parameters, we consider instead flattening the lower triangle of $\mathbf{M}_T \mathbf{M}_T^{\mathsf{T}} \in \mathbb{R}^{D_T \times D_T}$. The policy network then takes the flattened and normalised parameters as representations of the belief $b$. For MAML and RL$^2$, we use the implementation provided by Beck et al. (2023a). For VariBAD, we implement our own, following official implementations in Zintgraf et al. (2021); Beck et al. (2023a). The reason we re-implement VariBAD is to use the standardised framework provided by Beck et al. (2023a), written in JAX (Bradbury et al., 2018). Our VariBAD implementation has experiment results matching that of Beck et al. (2023a). We use the tuned hyperparameters from official implementations in MAML, RL$^2$ and VariBAD. The table of all related hyperparameters of GLiBRL is shown in Appendix A.10.

## 5.2 COMPARISONS AMONG GLiBRL, VARIBAD, MAML AND RL$^2$

We demonstrate the results in Figure 1, which shows the testing success rates with related to training steps. The success rates are averaged across 5 testing tasks (with 10 seeds per experiment), and per-task success rates are shown in Appendix A.5 and Appendix A.6. Overall, GLiBRL substantially

---

[5]For example, PEARL barely succeeded ($< 3\%$) in ML10 with 1e8 steps, see Figure 17 in Yu et al. (2021).

Table 1: Maximal testing success rate of GLiBRL, SDVT, TrMRL and ECET. Results from SDVT, TrMRL and ECET are taken as reported. Results of TrMRL are from ECET, as they are not reported in (Melo, 2022). $\dagger$: results with $5e7$ training steps - corresponding GLiBRL results with $5e7$ steps are bracketed. Best results are in **bold**.

| ML10 | | | | ML45 | | | |
|---|---|---|---|---|---|---|---|
| **GLiBRL** | **SDVT** | **TrMRL** | **ECET** | **GLiBRL** | **SDVT** | **TrMRL**$^\dagger$ | **ECET**$^\dagger$ |
| **0.25** | 0.19 | 0.14 | 0.18 | **0.45** | 0.20 | 0.23 (0.44) | 0.38 (0.44) |

outperforms one of the state-of-the-art deep BRL methods, VariBAD, and shows the lowest variance in both ML10 and ML45 benchmarks.

**GLiBRL and VariBAD:** The main comparison is between GLiBRL and VariBAD as they are both BRL methods. In both benchmarks, we can see substantial improvement, by up to $2.7\times$ in ML10, using GLiBRL. We also noticed an interesting decreasing trend of VariBAD in the ML45 benchmark. We suspect that the more number of training tasks leads to increased difficulty of learning meaningful and distinctive latent representations, partly because of the ELBO objective in VariBAD. By design, GLiBRL avoids the use of ELBO, hence achieving better and more stable performance. In Appendix A.7, we show that VariBAD suffers from posterior collapse even in the simpler benchmark ML10, while GLiBRL learns meaningful task representations.

**GLiBRL, MAML and RL$^2$:** GLiBRL achieves consistently higher success rates than MAML and RL$^2$ in the more complex ML45 benchmark. Notably, GLiBRL also admits low variance which can be inferred from the tightest CI. In Appendix A.9, we show that GLiBRL reveals even **higher** success rates (**29%**) in ML10 when setting $D_T = 8$, at the cost of slightly higher predictive error.

### 5.3 COMPARISONS AMONG GLiBRL, SDVT, TRMRL AND ECET

Aside from comparisons against standard baselines in deep BRL and Meta-RL, we compare the success rate of GLiBRL in both ML10 / ML45 to more recent work, namely SDVT (Lee et al., 2023), TrMRL (Melo, 2022) and ECET (Shala et al., 2025).

Table 1 demonstrates the detailed results. We can observe that GLiBRL outperforms all of these recent deep BRL / Meta-RL methods. Comparing against computationally heavy Transformer-based models TrMRL and ECET, GLiBRL is more performant and lightweight ($\sim 200$K parameters) hence applicable to, for example, mobile robots with lower-end computing resources.

Having outperformed most of the recent or state-of-the-art methods already, GLiBRL learns policies from PPO that is not fully revealing its potentials. Model-based policy learners that cannot be applied to black-box / PPG-based Meta-RL methods are expected to further improve the sample efficiency and performance of GLiBRL.

### 5.4 ABLATION STUDIES

GLiBRL can be viewed as a generalised deep BRL version of ALPaCA, as GLiBRL performs Bayesian inference on model noises $T_\sigma, R_\sigma$, while ALPaCA simply assumes $T_\sigma = \boldsymbol{\Sigma}_T, R_\sigma = \boldsymbol{\Sigma}_R$ are fully known a priori. Under the assumption of ALPaCA, Equation 14 reduces to (see Appendix A.4)

$$\log_{\phi_T, \phi_R}(\mathcal{C}) = -\frac{1}{2} \sum_{i=1}^{M} D_S \log |\boldsymbol{\Xi}'_T| - \text{Tr}(\boldsymbol{\Sigma}_T^{-1} \mathbf{M}_T'^{\text{T}} \boldsymbol{\Xi}'_T \mathbf{M}_T')$$

$$-\frac{1}{2} \sum_{i=1}^{M} \log |\boldsymbol{\Xi}'_R| - \text{Tr}(\boldsymbol{\Sigma}_R^{-1} \mathbf{M}_R'^{\text{T}} \boldsymbol{\Xi}'_R \mathbf{M}_R') + \text{const.} \quad (18)$$

We studied on whether inferring on model noises is necessary for learning accurate transition and reward models. GLiBRL and its variant without noise inference (GLiBRL_wo_NI) are tested with

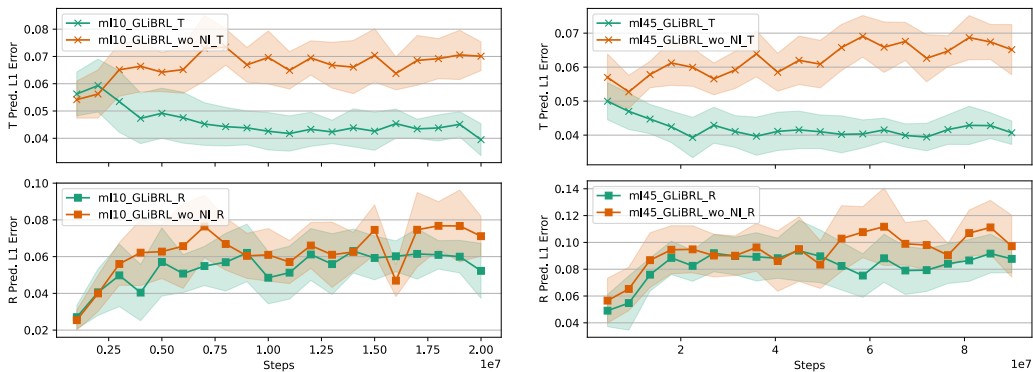

Figure 2: IQM and $95\%$ CI of errors in transition and reward predictions, comparing GLiBRL and GLiBRL_wo_NI. Up: Transitions; Bottom: Rewards; Left: ML10; Right: ML45.

identical hyperparameters on both ML10 and ML45 benchmarks [6]. The metric being evaluated are $L_1$ norms of prediction errors in both transitions (defined as $|\mathbf{S}' - \mathbf{C}_T T_\mu|_1$) and rewards (defined as $|\mathbf{r} - \mathbf{C}_R R_\mu|_1$). [7] The results are shown in Figure 2.

Overall, with noise inference, GLiBRL admits lower prediction errors in both transitions and rewards, compared to GLiBRL_wo_NI assuming known noises. Both methods become increasingly erroneous in reward predictions with training steps. This is expected, as more steps result in higher success rates, hence increased magnitude of rewards and errors. However, the increasing trend of transition errors of GLiBRL_wo_NI is abnormal, as magnitudes of states are rather bounded and less relevant with the success rate, compared to that of rewards. In Equation 18, the term governing the fit of the transition model $\mathrm{Tr}(\mathbf{\Sigma}_T^{-1}\mathbf{M}_T'^{\mathsf{T}}\mathbf{\Xi}_T'\mathbf{M}_T')$ has a fixed learning rate from the fixed $\mathbf{\Sigma}_T$, leading to continual unstable / underfit behaviours if the learning rate is too big / too small. On the contrary, in GLiBRL, the model fit term $\nu_T' \log|\frac{1}{2}\mathbf{\Omega}_T'|$ has dynamic learning rates from dynamic $\nu_T'$. This enables self-adaptive and effective model learning, hence the expected decreasing trend in Figure 2. The lower prediction error of GLiBRL allows better integrations with model-based methods using imaginary samples, the quality of which depending highly on the accuracy of the prediction.

## 6 CONCLUSION

We propose GLiBRL, a novel deep BRL method that enables fully tractable inference on the task parameters and efficient learning of basis functions with ELBO-free optimisation. Instead of assuming known noises of models, GLiBRL performs Bayesian inference, which has been shown empirically to reduce the error of prediction in both transition and reward models. The results on challenging MetaWorld ML10 and ML45 benchmarks demonstrate a substantial improvement compared to one of the state-of-the-art deep BRL methods, VariBAD. Low-variance and decent performance of GLiBRL can also be inferred from its comparisons against representative or recent deep BRL / Meta-RL methods, including MAML, RL[2], SDVT, TrMRL and ECET.

Multiple directions of future work arise naturally from the formulation of GLiBRL, with the most interesting one being model-based methods. As GLiBRL is capable of learning accurate transition and reward models, model-based methods can be applied easily for improved sample efficiency and performance. However, model-based methods usually require frequent sampling from the learnt models, revealing limitations in GLiBRL, as sampling from Wishart distributions can be slow. Another exciting direction is, if we prefer model-free methods, to seek a better way of utilising the task parameters in the policy network. In the paper, we simply feed the policy network with normalised means of task parameters. A naive normalisation of the parameters may confuse the policy network, and the use of means only loses uncertainty information from covariances.

---

[6]They also have the same initial noises. $(\nu_T\mathbf{\Omega}_T)^{-1} = \mathbf{\Sigma}_T = 0.025 \cdot \mathbf{I}$ and $(\nu_R\mathbf{\Omega}_R)^{-1} = \mathbf{\Sigma}_R = 0.5$.

[7]Comparisons of success rates are not included, as there is no obvious difference in IQMs or CIs.

ETHICS STATEMENT

This work does not raise any questions regarding the Code of Ethics.

REPRODUCIBILITY STATEMENT

All experiments can be reproduced with the source code provided in the supplementary materials.

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

# A  APPENDIX

## USE OF LLMS

This work does not use LLMs in any significant ways.

## A.1  EVIDENCE LOWER BOUND

The evidence lower bound in Equation 4 is derived as follows

$$\log p_{\phi_T,\phi_R}(\mathcal{C}) = \sum_{i=1}^{M} \log \iint p(\mathcal{C}^{\mathcal{M}_i}, \theta_T, \theta_R) \frac{q(\theta_T, \theta_R | \mathcal{C}^{\mathcal{M}_i})}{q(\theta_T, \theta_R | \mathcal{C}^{\mathcal{M}_i})} d\theta_T d\theta_R$$

$$\geq \sum_{i=1}^{M} \mathbb{E}_q \left[ \log \frac{p(\theta_T, \theta_R)}{q(\theta_T, \theta_R | \mathcal{C}^{\mathcal{M}_i})} + \log p(\mathcal{C}^{\mathcal{M}_i} | \theta_T, \theta_R) \right]$$

$$= \sum_{i=1}^{M} \mathbb{E}_q \left[ \log p(\mathcal{C}^{\mathcal{M}_i} | \theta_T, \theta_R) \right] - D_{KL} \left( q(\theta_T, \theta_R | \mathcal{C}^{\mathcal{M}_i}) \| p(\theta_T, \theta_R) \right)$$

## A.2 Normal-Wishart-Normal Conjugacy

Given Equation 9

$$p(\theta_T) = p(T_\mu, T_\sigma) = \mathcal{MN}(T_\mu | \mathbf{M}_T \in \mathbb{R}^{D_T \times D_S}, \mathbf{\Xi}_T^{-1} \in \mathbb{R}^{D_T \times D_T}, T_\sigma)$$
$$\cdot \mathcal{W}(T_\sigma^{-1} | \mathbf{\Omega}_T^{-1} \in \mathbb{R}^{D_S \times D_S}, \nu_T \in \mathbb{R}_{++}) \tag{9}$$

and Equation 5

$$p_{\phi_T}(\mathbf{S}_i' | \mathbf{S}_i, \mathbf{A}_i, \theta_T) = \mathcal{MN}(\mathbf{S}_i' | \mathbf{C}_T T_\mu, \mathbf{I}_N, T_\sigma) \tag{5}$$

We prove Equation 11

$$p_{\phi_T}(\theta_T | \mathbf{S}_i, \mathbf{A}_i, \mathbf{S}_i') = \mathcal{MN}(T_\mu | \mathbf{M}_T', \mathbf{\Xi}_T'^{-1}, T_\sigma) \cdot \mathcal{W}(T_\sigma^{-1} | \mathbf{\Omega}_T'^{-1}, \nu_T') \tag{11}$$

where

$$\mathbf{M}_T' = \mathbf{\Xi}_T'^{-1} \left[ \mathbf{C}_T^{\mathrm{T}} \mathbf{S}' + \mathbf{\Xi}_T \mathbf{M}_T \right]$$
$$\mathbf{\Xi}_T' = \mathbf{C}_T^{\mathrm{T}} \mathbf{C}_T + \mathbf{\Xi}_T$$
$$\mathbf{\Omega}_T' = \mathbf{\Omega}_T + \mathbf{S}'^{\mathrm{T}} \mathbf{S}' \tag{13}$$
$$+ \mathbf{M}_T^{\mathrm{T}} \mathbf{\Xi}_T \mathbf{M}_T - \mathbf{M}_T'^{\mathrm{T}} \mathbf{\Xi}_T' \mathbf{M}_T'$$
$$\nu_T' = \nu_T + N$$

Proof:

The density of the prior distribution $p(\theta_T)$ is

$$p(\theta_T) = \frac{|\mathbf{\Xi}_T|^{D_S/2}}{\sqrt{(2\pi)^{D_T D_S} |T_\sigma|^{D_T/2}}} \cdot \exp\left[ -\frac{1}{2} \operatorname{Tr}\left( T_\sigma^{-1} (T_\mu - \mathbf{M}_T)^{\mathrm{T}} \mathbf{\Xi}_T (T_\mu - \mathbf{M}_T) \right) \right]$$
$$\cdot \sqrt{\frac{|\mathbf{\Omega}_T|^{\nu_T}}{2^{\nu_T D_S}}} \frac{|T_\sigma|^{(1-D_S-\nu_T)/2}}{\Gamma_{D_S}(\nu_T/2)} \cdot \exp\left[ -\frac{1}{2} \operatorname{Tr}\left( \mathbf{\Omega}_T T_\sigma^{-1} \right) \right] \tag{19}$$
$$\propto |T_\sigma|^{(1-D_S-\nu_T-D_T)/2} \cdot \exp\left\{ -\frac{1}{2} \operatorname{Tr}\left[ T_\sigma^{-1} \left[ (T_\mu - \mathbf{M}_T)^{\mathrm{T}} \mathbf{\Xi}_T (T_\mu - \mathbf{M}_T) + \mathbf{\Omega}_T \right] \right] \right\} \tag{20}$$

where from Equation 19 to Equation 20 we treat multiplicative parameters irrelevant to $\theta_T$ as constants. The joint density of $p_{\phi_T}(\theta_T, \mathbf{S}_i' | \mathbf{S}_i, \mathbf{A}_i) = p(\theta_T) \cdot p_{\phi_T}(\mathbf{S}_i' | \mathbf{S}_i, \mathbf{A}_i, \theta_T)$ is hence

$$\frac{|\mathbf{\Xi}_T|^{D_S/2}}{\sqrt{(2\pi)^{D_T D_S} |T_\sigma|^{D_T/2}}} \cdot \exp\left[ -\frac{1}{2} \operatorname{Tr}\left( T_\sigma^{-1} (T_\mu - \mathbf{M}_T)^{\mathrm{T}} \mathbf{\Xi}_T (T_\mu - \mathbf{M}_T) \right) \right]$$
$$\cdot \sqrt{\frac{|\mathbf{\Omega}_T|^{\nu_T}}{2^{\nu_T D_S}}} \frac{|T_\sigma|^{(1-D_S-\nu_T)/2}}{\Gamma_{D_S}(\nu_T/2)} \cdot \exp\left[ -\frac{1}{2} \operatorname{Tr}\left( \mathbf{\Omega}_T T_\sigma^{-1} \right) \right] \tag{21}$$
$$\cdot \frac{1}{\sqrt{(2\pi)^{N D_S} |T_\sigma|^{N/2}}} \cdot \exp\left[ -\frac{1}{2} \operatorname{Tr}\left( T_\sigma^{-1} (\mathbf{S}_i' - \mathbf{C}_T T_\mu)^{\mathrm{T}} (\mathbf{S}_i' - \mathbf{C}_T T_\mu) \right) \right]$$

$$\propto |T_\sigma|^{(1-D_S-\nu_T-N-D_T)/2} \cdot \exp\left\{ -\frac{1}{2} \operatorname{Tr}\left[ T_\sigma^{-1} \left[ (T_\mu - \mathbf{M}_T)^{\mathrm{T}} \mathbf{\Xi}_T (T_\mu - \mathbf{M}_T) \right. \right. \right.$$
$$\left. \left. \left. + (\mathbf{S}_i' - \mathbf{C}_T T_\mu)^{\mathrm{T}} (\mathbf{S}_i' - \mathbf{C}_T T_\mu) + \mathbf{\Omega}_T \right] \right] \right\} \tag{22}$$

Matching the second-order then the first-order term with related to $T_\mu$, we can rewrite

$$22 = |T_\sigma|^{(1-D_S-\nu_T'-D_T)/2} \cdot \exp\left\{ -\frac{1}{2} \operatorname{Tr}\left[ T_\sigma^{-1} \left[ (T_\mu - \mathbf{M}_T')^{\mathrm{T}} \mathbf{\Xi}_T' (T_\mu - \mathbf{M}_T') + \mathbf{\Omega}_T' \right] \right] \right\} \tag{23}$$

We can find Equation 20 and Equation 23 match exactly, indicating the Normal-Wishart-Normal conjugacy. Note, we just use the posterior update of $p(\theta_T)$ as an example. The exact same proof applies to the posterior update of $p(\theta_R)$ as well. Such conjugacy allows exact posterior update and marginal likelihood, enabling efficient learning.

## A.3 Marginal Log-likelihood of Normal-Wishart-Normal

We prove Equation 14 has the following closed form

$$\log p_{\phi_T, \phi_R}(\mathcal{C}) = -\frac{1}{2} \sum_{i=1}^{M} D_S \log |\mathbf{\Xi}'_T| + \nu'_T \log |\frac{1}{2}\mathbf{\Omega}'_T| + \log |\mathbf{\Xi}'_R| + \nu'_R \log |\frac{1}{2}\mathbf{\Omega}'_R| + \text{const.} \quad (14)$$

Proof:

Consider

$$p_{\phi_T}(\mathbf{S}'_i|\mathbf{S}_i, \mathbf{A}_i) = \frac{p_{\phi_T}(\theta_T, \mathbf{S}'_i|\mathbf{S}_i, \mathbf{A})}{p_{\phi_T}(\theta_T|\mathbf{S}_i, \mathbf{A}_i, \mathbf{S}'_i)} \quad (24)$$

From Appendix A.2, we know the numerator is

$$\frac{|\mathbf{\Xi}_T|^{D_S/2}|\mathbf{\Omega}_T|^{\nu_T/2}}{2^{\nu_T D_S/2} \cdot (2\pi)^{D_S(D_T+N)/2} \cdot \Gamma_{D_S}(\nu_T/2)} \cdot \text{Equation 23} \quad (25)$$

The denominator is

$$\frac{|\mathbf{\Xi}'_T|^{D_S/2}|\mathbf{\Omega}'_T|^{\nu'_T/2}}{2^{\nu'_T D_S/2} \cdot (2\pi)^{D_S D_T/2} \cdot \Gamma_{D_S}(\nu'_T/2)} \cdot \text{Equation 23} \quad (26)$$

Hence,

$$p_{\phi_T}(\mathbf{S}'_i|\mathbf{S}_i, \mathbf{A}_i) = \frac{1}{(2\pi)^{D_S N/2}} \cdot \frac{|\mathbf{\Xi}_T|^{D_S/2}|\frac{1}{2}\mathbf{\Omega}_T|^{\nu_T/2} \cdot \Gamma_{D_S}(\nu'_T/2)}{|\mathbf{\Xi}'_T|^{D_S/2}|\frac{1}{2}\mathbf{\Omega}'_T|^{\nu'_T/2} \cdot \Gamma_{D_S}(\nu_T/2)} \quad (27)$$

Similarly,

$$p_{\phi_R}(\mathbf{r}_i|\mathbf{S}_i, \mathbf{A}_i, \mathbf{S}'_i) = \frac{1}{(2\pi)^{N/2}} \cdot \frac{|\mathbf{\Xi}_R|^{1/2}|\frac{1}{2}\mathbf{\Omega}_R|^{\nu_R/2} \cdot \Gamma(\nu'_R/2)}{|\mathbf{\Xi}'_R|^{1/2}|\frac{1}{2}\mathbf{\Omega}'_R|^{\nu'_R/2} \cdot \Gamma(\nu_R/2)} \quad (28)$$

Note that

$$p_{\phi_T, \phi_R}(\mathcal{C}^{\mathcal{M}_i}) = p_{\phi_T}(\mathbf{S}'_i|\mathbf{S}_i, \mathbf{A}_i) p_{\phi_R}(\mathbf{r}_i|\mathbf{S}_i, \mathbf{A}_i, \mathbf{S}'_i) p(\mathbf{S}_i, \mathbf{A}_i) \quad (29)$$

By taking the logarithm, and note the independence of $p(\mathbf{S}_i, \mathbf{A}_i)$ with related to $\phi_T, \phi_R$,

$$\log p_{\phi_T, \phi_R}(\mathcal{C}^{\mathcal{M}_i}) = \log p_{\phi_T}(\mathbf{S}'_i|\mathbf{S}_i, \mathbf{A}_i) + \log p_{\phi_R}(\mathbf{r}_i|\mathbf{S}_i, \mathbf{A}_i, \mathbf{S}'_i) + \text{const.} \quad (30)$$

Sum the above equation on both sides with related to $i$, then we have Equation 14.

## A.4 Marginal Log-likelihood of Normal-Normal

We prove Equation 18 has the following form

$$\log_{\phi_T, \phi_R}(\mathcal{C}) = -\frac{1}{2} \sum_{i=1}^{M} D_S \log |\mathbf{\Xi}'_T| - \mathrm{Tr}(\mathbf{\Sigma}_T^{-1} \mathbf{M}'^{\mathrm{T}}_T \mathbf{\Xi}'_T \mathbf{M}'_T)$$

$$-\frac{1}{2} \sum_{i=1}^{M} \log |\mathbf{\Xi}'_R| - \mathrm{Tr}(\mathbf{\Sigma}_R^{-1} \mathbf{M}'^{\mathrm{T}}_R \mathbf{\Xi}'_R \mathbf{M}'_R) + \mathrm{const.} \quad (18)$$

Proof:

The distributions without inferring on the noise are listed as follows:

**Likelihood:**

$$p_{\phi_T}(\mathbf{S}'_i | \mathbf{S}_i, \mathbf{A}_i, \theta_T) = \mathcal{MN}\left(\mathbf{S}'_i | \mathbf{C}_T T_\mu, \mathbf{I}_N, \mathbf{\Sigma}_T \in \mathbb{R}^{D_S \times D_S}\right) \quad (31)$$

$$p_{\phi_R}(\mathbf{r}_i | \mathbf{S}_i, \mathbf{A}_i, \mathbf{S}'_i, \theta_R) = \mathcal{MN}\left(\mathbf{r}_i | \mathbf{C}_R R_\mu, \mathbf{I}_N, \mathbf{\Sigma}_R \in \mathbb{R}^{1 \times 1}\right) \quad (32)$$

**Prior:**

$$p(\theta_T) = p(T_\mu) = \mathcal{MN}(T_\mu | \mathbf{M}_T \in \mathbb{R}^{D_T \times D_S}, \mathbf{\Xi}_T^{-1} \in \mathbb{R}^{D_T \times D_T}, \mathbf{\Sigma}_T) \quad (33)$$

$$p(\theta_R) = p(R_\mu) = \mathcal{MN}(R_\mu | \mathbf{M}_R \in \mathbb{R}^{D_R \times 1}, \mathbf{\Xi}_R^{-1} \in \mathbb{R}^{D_R \times D_R}, \mathbf{\Sigma}_R) \quad (34)$$

**Posterior:**

$$p_{\phi_T}(\theta_T | \mathbf{S}_i, \mathbf{A}_i, \mathbf{S}'_i) = \mathcal{MN}(T_\mu | \mathbf{M}'_T, \mathbf{\Xi}'^{-1}_T, \mathbf{\Sigma}_T) \quad (35)$$

$$p_{\phi_R}(\theta_R | \mathbf{S}_i, \mathbf{A}_i, \mathbf{S}'_i, \mathbf{r}) = \mathcal{MN}(R_\mu | \mathbf{M}'_R, \mathbf{\Xi}'^{-1}_R, \mathbf{\Sigma}_R) \quad (36)$$

where

$$\mathbf{M}'_T = \mathbf{\Xi}'_T{}^{-1}\left[\mathbf{C}_T^{\mathrm{T}} \mathbf{S}' + \mathbf{\Xi}_T \mathbf{M}_T\right] \qquad \mathbf{M}'_R = \mathbf{\Xi}'_R{}^{-1}\left[\mathbf{C}_R^{\mathrm{T}} \mathbf{r} + \mathbf{\Xi}_R \mathbf{M}_R\right]$$
$$\mathbf{\Xi}'_T = \mathbf{C}_T^{\mathrm{T}} \mathbf{C}_T + \mathbf{\Xi}_T \qquad\qquad \mathbf{\Xi}'_R = \mathbf{C}_R^{\mathrm{T}} \mathbf{C}_R + \mathbf{\Xi}_R \quad (37)$$

Similar to Appendix A.3,

$$p_{\phi_T}(\mathbf{S}'_i | \mathbf{S}_i, \mathbf{A}_i) = \frac{p_{\phi_T}(\theta_T, \mathbf{S}'_i | \mathbf{S}_i, \mathbf{A})}{p_{\phi_T}(\theta_T | \mathbf{S}_i, \mathbf{A}_i, \mathbf{S}'_i)} \quad (38)$$

As

$$p_{\phi_T}(\theta_T, \mathbf{S}'_i | \mathbf{S}_i, \mathbf{A}) = \frac{|\mathbf{\Xi}_T|^{D_S/2} |\mathbf{\Sigma}_T|^{-D_T/2}}{\sqrt{(2\pi)^{D_T D_S}}} \cdot \exp\left[-\frac{1}{2} \mathrm{Tr}\left(\mathbf{\Sigma}_T^{-1}(T_\mu - \mathbf{M}_T)^{\mathrm{T}} \mathbf{\Xi}_T (T_\mu - \mathbf{M}_T)\right)\right]$$

$$\cdot \frac{|\mathbf{\Sigma}_T|^{-N/2}}{\sqrt{(2\pi)^{N D_S}}} \cdot \exp\left[-\frac{1}{2} \mathrm{Tr}\left(\mathbf{\Sigma}_T^{-1}(\mathbf{S}'_i - \mathbf{C}_T T_\mu)^{\mathrm{T}}(\mathbf{S}'_i - \mathbf{C}_T T_\mu)\right)\right]$$

$$(39)$$

And

$$p_{\phi_T}(\theta_T | \mathbf{S}_i, \mathbf{A}_i, \mathbf{S}'_i) = \frac{|\mathbf{\Xi}'_T|^{D_S/2} |\mathbf{\Sigma}_T|^{-D_T/2}}{\sqrt{(2\pi)^{D_T D_S}}} \cdot \exp\left[-\frac{1}{2} \mathrm{Tr}\left(\mathbf{\Sigma}_T^{-1}(T_\mu - \mathbf{M}'_T)^{\mathrm{T}} \mathbf{\Xi}'_T (T_\mu - \mathbf{M}'_T)\right)\right]$$

$$(40)$$

Hence

$$p_{\phi_T}(\mathbf{S}'_i | \mathbf{S}_i, \mathbf{A}_i) \propto |\mathbf{\Xi}'_T|^{-D_S/2} \cdot \exp\left[-\frac{1}{2} \mathrm{Tr}\left(-\mathbf{M}'^{\mathrm{T}}_T \mathbf{\Xi}'_T \mathbf{M}'_T\right)\right] \quad (41)$$

$$\log p_{\phi_T}(\mathbf{S}'_i | \mathbf{S}_i, \mathbf{A}_i) = -\frac{D_S}{2} |\mathbf{\Xi}'_T| + \frac{1}{2} \mathrm{Tr}\left(\mathbf{M}'^{\mathrm{T}}_T \mathbf{\Xi}'_T \mathbf{M}'_T\right) + \mathrm{const.} \quad (42)$$

Similarly,

$$\log p_{\phi_R}(\mathbf{r}_i|\mathbf{S}_i, \mathbf{A}_i, \mathbf{S}'_i) = -\frac{1}{2}|\mathbf{\Xi}'_R| + \frac{1}{2}\operatorname{Tr}\left(\mathbf{M}'^{\mathrm{T}}_R \mathbf{\Xi}'_R \mathbf{M}'_R\right) + \text{const.} \tag{43}$$

Following Appendix A.3,

$$\log_{\phi_T,\phi_R}(\mathcal{C}) = -\frac{1}{2}\sum_{i=1}^{M} D_S \log|\mathbf{\Xi}'_T| - \operatorname{Tr}(\mathbf{\Sigma}_T^{-1}\mathbf{M}'^{\mathrm{T}}_T \mathbf{\Xi}'_T \mathbf{M}'_T)$$

$$-\frac{1}{2}\sum_{i=1}^{M} \log|\mathbf{\Xi}'_R| - \operatorname{Tr}(\mathbf{\Sigma}_R^{-1}\mathbf{M}'^{\mathrm{T}}_R \mathbf{\Xi}'_R \mathbf{M}'_R) + \text{const.} \tag{18}$$

### A.5 ML10 Success Rate Comparisons Per-task

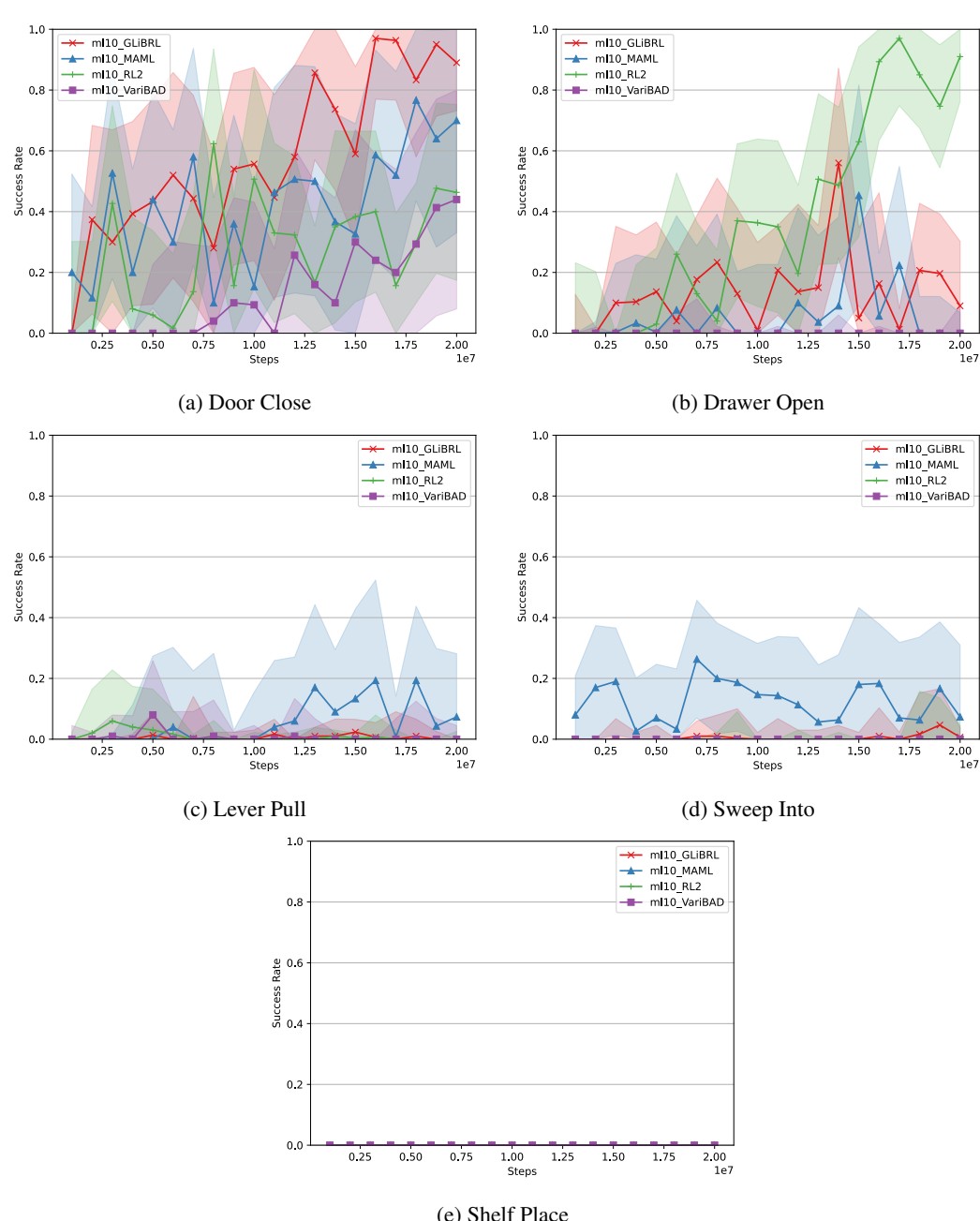

Figure 3: IQM and 95% CI of success rate for each testing scenario in ML10. The Shelf Place scenario is challenging as none of the method can achieve a single success.

A.6   ML45 SUCCESS RATE COMPARISONS PER-TASK

(a) Door Lock

(b) Door Unlock

(c) Hand Insert

(d) Bin Picking

(e) Box Close

Figure 4: IQM and 95% CI of success rate for each testing scenario in ML45. GLiBRL achieves nearly 100% testing success rates in both Door Lock and Door Unlock scenarios.

## A.7 POSTERIOR COLLAPSE IN VARIBAD

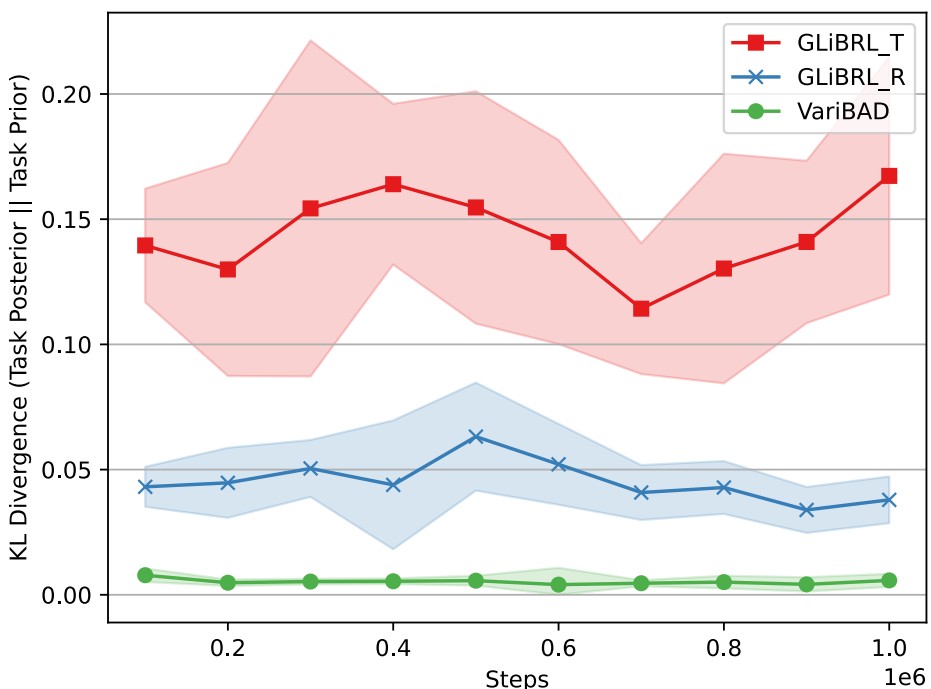

To verify if VariBAD can learn meaningful task representations, we check the IQM and 95% CI of the expected KL-divergence between task posteriors and task priors $\mathbb{E}\left[D_{KL}\left(q(\theta_T, \theta_R | \mathcal{C}_{t+1}) \,||\, q(\theta_T, \theta_R | \mathcal{C}_t)\right)\right]$, out of 10 runs in the ML10 benchmark, where $\mathcal{C}_{t+1} = \mathcal{C}_t \cup c_{t+1}$ updates the set of contexts $\mathcal{C}_t$ with the context $c_{t+1}$ at time step $t + 1$. VariBAD uses a single latent variable to model both transitions and rewards, hence contributing to only one line in the above figure.

Intuitively, if the expected divergence is close to 0, the majority of posterior updates has collapsed to priors, meaning barely any meaningful task representation has been learnt. Clearly from the above figure, VariBAD fails to learn meaningful representations, while GLiBRL demonstrates obvious divergence between posteriors and priors.

## A.8    REGULARISATION IN THE MARGINAL LOG-LIKELIHOOD

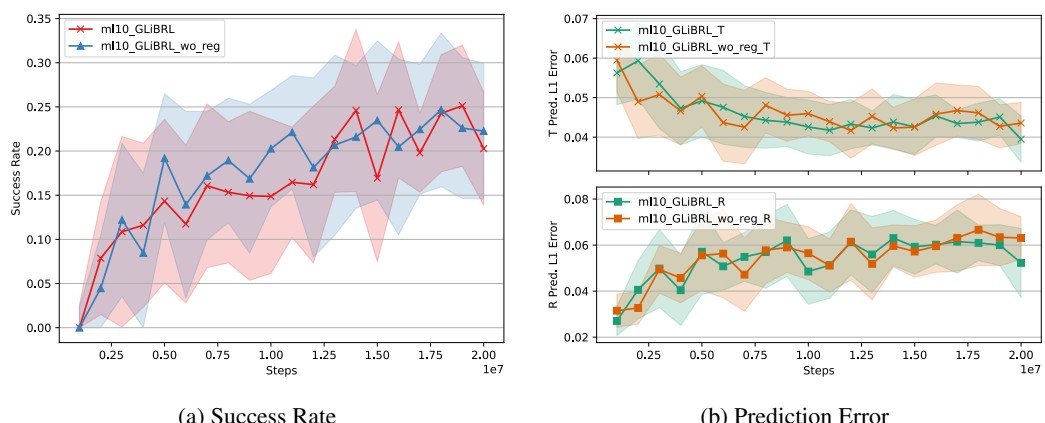

(a) Success Rate                                    (b) Prediction Error

Running GLiBRL and its variant without regularisation in the ML10 benchmark, we can tell the regularisations do not change the overall trend in both success rate and prediction error. However, as the number of training step increases, GLiBRL begins to show in (a): higher success rates with narrower CI; in (b): lower prediction error in both transitions and rewards.

## A.9 SENSITIVITY OF LATENT TASK DIMENSIONS

To demonstrate the sensitivity of GLiBRL with related to latent task dimensions $D_T$ and $D_R$, we perform hyperparameter search in ML10 on (1) $D_T = \{4, 8, 16, 32\}$ while fixing $D_R = 256$ and (2) $D_R = \{32, 64, 128, 256, 512\}$ while fixing $D_T = 16$. We compare on (1) the success rate, (2) the predictive error in transitions, and (3) the predictive error in rewards.

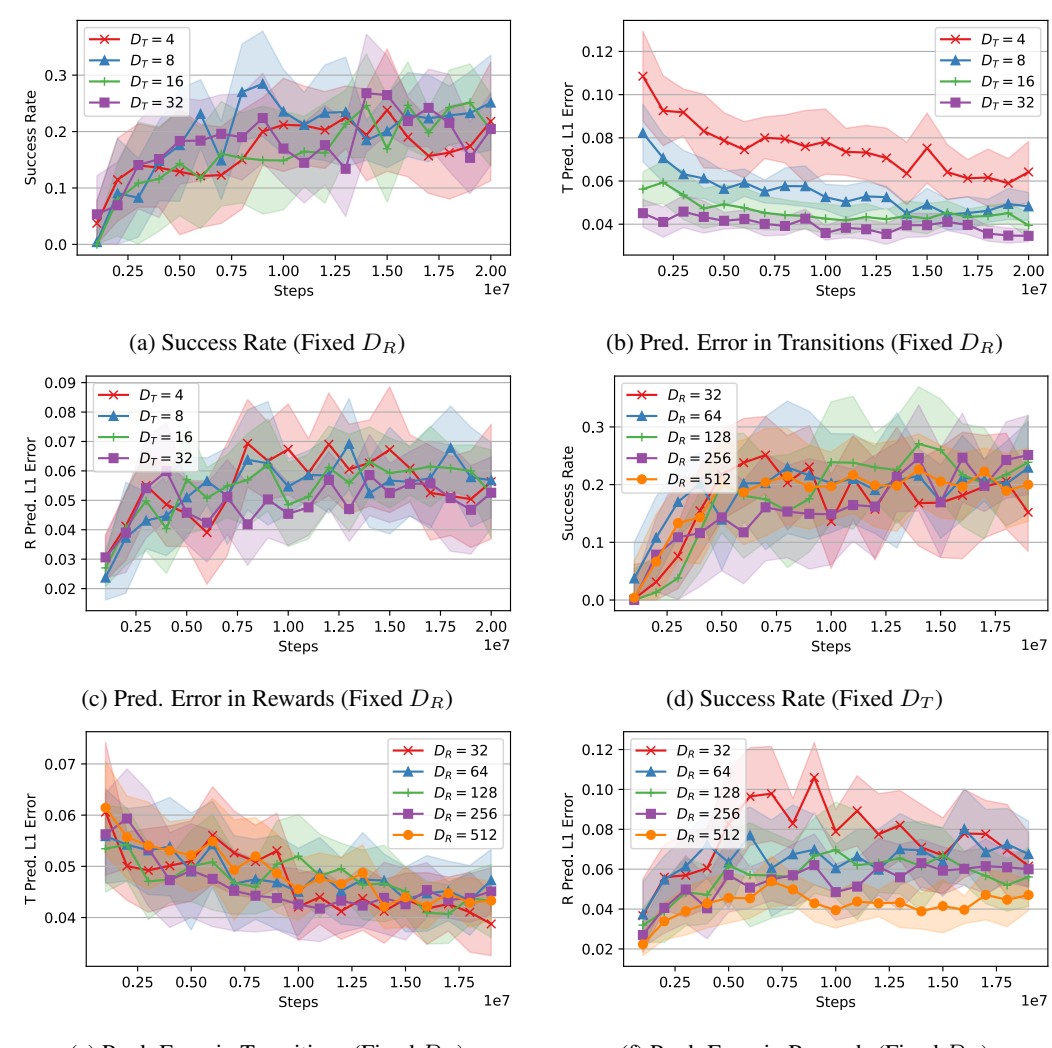

(a) Success Rate (Fixed $D_R$)

(b) Pred. Error in Transitions (Fixed $D_R$)

(c) Pred. Error in Rewards (Fixed $D_R$)

(d) Success Rate (Fixed $D_T$)

(e) Pred. Error in Transitions (Fixed $D_T$)

(f) Pred. Error in Rewards (Fixed $D_T$)

In general, GLiBRL shows stable success rates with related to latent task dimensions $D_T, D_R$, which can be inferred from Figure (a) and (d). We have observed that GLiBRL achieves state-of-the-art performance on ML10 (**29**% success rate) with $D_T = 8$, with the cost of predictive errors. Figure (b) and (f) indicate that as $D_T, D_R$ grows, the corresponding error reduces, offering an efficiency-accuracy trade-off. Figure (c) and (e) are sanity checks that have confirmed transitions do not affect rewards, and vice versa. Overall, $D_R$ are set to be much larger than $D_T$, as reward functions are generally much harder to learn, compared to transition function.

## A.10 HYPERPARAMETERS, RUNTIME AND MEMORY

We list all hyperparameters of GLiBRL in the following table. We use the same hyperparameters for both ML10 and ML45.

| Name | Value | Name | Value |
|---|---|---|---|
| policy_learner | PPO | feat_out_activation | True |
| policy_layers | [256, 256] | t_mix_layers | [64, 32] |
| a_feat_out_activation | True | t_mix_layernorm | True |
| policy_activation | Tanh | t_mix_out_activation | False |
| policy_optimiser | Adam | r_mix_layers | [128, 64] |
| policy_lr | 5e-4 | r_mix_layernorm | True |
| policy_opt_max_norm | 1 | r_mix_out_activation | False |
| policy_weight_init | Xavier | t_reg_coef | 5e-3 |
| policy_bias_init | 0 | r_reg_coef | 1e-3 |
| policy_log_std_min | 1e-6 | model_activation | ReLU |
| policy_log_std_max | 2 | model_optimiser | Adam |
| policy_grad_epochs | 10 | model_lr | 2e-4 |
| policy_grad_steps | 20 | model_opt_max_norm | None |
| ppo_clip_eps | 0.5 | model_grad_epochs | 1 |
| ppo_gamma | 0.99 | model_grad_steps | 20 |
| ppo_gae_lambda | 0.95 | init_mt | zeros |
| ppo_entropy_coef | 5e-3 | init_mr | zeros |
| s_feat_layers | [64, 32] | init_xit | ones |
| s_feat_outdim | 32 | init_xir | ones |
| s_feat_layernorm | False | init_omegat | ones |
| a_feat_layers | [32, 16] | init_omegar | ones |
| a_feat_outdim | 16 | init_nut | 40 |
| a_feat_layernorm | False | init_nur | 2 |

GLiBRL is rather efficient in both time and memory. Although all of our experiments are run using A100, we have tested that GLiBRL can run fast on much lower-end GPUs with 8GB memory, such as RTX 3070, with each run costing less than 2 hours. The runtime does not vary too much with changes in $D_T$ and $D_R$, due to the quadratic online inference complexity.

