# OpenReview forum: "Generalised Linear Models in Bayesian RL with Learnable Basis Functions"
_ICLR.cc/2026/Conference — Submitted to ICLR 2026_

### Official Review · Reviewer_qycn · 2025-10-30

**Soundness:** 3
**Presentation:** 3
**Contribution:** 3
**Rating:** 6
**Confidence:** 3

**Summary:**

The work proposes a novel Bayesian RL method which avoids the difficulty of optimization of the ELBO. This is achieved by using generalized linear models. The work begins by thoroughly motivating why BRL methods commonly require optimizing the ELBO before highlighting how the linear assumption enables them to avoid this issue at all. The resulting method provides a tractable, permutation-invariant posterior which allows for closed form marginal likelyhood updating.
The empirical evaluation highlights that the proposed GLiBRL approach has the capability of outperforming BRL/meta-RL baselines in the challenging metaworld setting.

Overall it seems like an interesting contribution to ICLR and I am leaning towards acceptance.

**Strengths:**

The work is very well motivated and is a good fit for the ICRL community. I believe that it could serve as the basis for engaging discussions in the subset of the RL community. The approach seems very novel and avoids a significant hurdle in existing BRL methods. Thus, this work likely will spawn future works that explore other approaches to avoid having to deal with ELBO when working with Bayesian RL.

I appreciate the depth of section 2.1 as it nicely sets up the following section in which the GLiBRL method is introduced.

The empirical evaluation seems meaningful though complementing the experiments with a simpler setting that requires less compute might help the community to quicker evaluate their own approaches against GLiBRL.

I also want to highlight that I am very appreciative of code being available already during the reviewing period.

**Weaknesses:**

Some implementation details and design decisions should be discussed in a bit more detail. For example, in line 359 the dimensionality of $\theta_T$ and $\theta_R$ are set but it is not clear how relevant this design decision is. Why did the reward get a much larger dimensionality than the transitions? Could that setting be an explanation for the results in Figure 2 being less decisive for the rewards than the transitions?

The results could be a bit stronger if other meta-RL methods would have been considered that work better on metaworld. E.g. the work by Melo (https://proceedings.mlr.press/v162/melo22a.html) or Shala et al. (https://openreview.net/forum?id=UENQuayzr1).
In Line 418 it is stated that GLiBRL outperforms the "state-of-the-art", though I'm doubtful that this is the case.

I'm happy to increase my score if my questions get clarified and the explanation for design decisions makes it clearer how difficult it is to setup GLiBRL.

**Questions:**

* For how many seeds are you reporting the results?
* How large is the overhead of sampling for a Wishart distribution with GLiBRL?
* How did you tune the hyperparameters of GLiBRL/How sensitive is GLiBRL with respect to hyperparameter choices?
* Why is Table 1 included? What does the max value tell us that is not shown from the final and average performances?

---

> ### Author Response · Authors · 2025-11-19
>
> Dear Reviewer qycn:
>
> Thank your for your encouraging feedback. We like to use this opportunity to address the concerns and clarify the questions.
>
> $\textbf{(W1)}$: *Some implementation details and design decisions should be discussed in a bit more detail ... Why did the reward get a much larger dimensionality than the transitions? Could that setting be an explanation for the results in Figure 2 being less decisive for the rewards than the transitions?*
>
> $\textbf{Re (W1)}$: We have included a sweep on $D_T$ and $D_R$ in Appendix A.9. The results indicate that GLiBRL is stable and remains performant with related to changes in $D_T$ and $D_R$. This also offers an efficiency-accuracy trade-off, i.e. $D_T$ and $D_R$ can be set larger if more accurate models are desired. $D_R$ is set to be much larger than $D_T$, as reward functions are generally harder to learn than transition functions. As for Figure 2, in A.9, we have shown that if we keep increasing $D_R$, we can better learn the reward function. We have also observed that with $D_T = 8$, GLiBRL has success rate of $\textbf{29}\%$, outperforming all the comparators.
>
> $\textbf{(W2)}$:*The results could be a bit stronger if other meta-RL methods would have been considered that work better on metaworld. E.g. the work by Melo or Shala et al.. In Line 418 it is stated that GLiBRL outperforms the "state-of-the-art", though I'm doubtful that this is the case.*
>
> $\textbf{Re (W2)}$: We really appreciate that you pointed out these more recent transformer-based Meta-RL methods. Despite their novelty in using transformer for Meta-RL, their learning efficiency and performance are still not as good compared to GLiBRL. We list the comparison in the table below (also in section 5.3 of the updated submission), including more recent work such as SDVT [Lee, 2023], TrMRL [Melo, 2022], ECET [Shala, 2025] and a work under review at ICLR 26, DME [Anonymous, 2025]:
>
> | Method                 | ML10 (2e7 steps) | ML45 (9e7 steps)                                                 |
> |------------------------|------------------|------------------------------------------------------------------|
> | **GLiBRL**             | **25%**          | **45%**                                                          |
> | SDVT [Lee, 2023]       | 19%              | 20%                                                              |
> | ECET [Shala, 2025]     | 18%              | 38% (5e7 steps) (GLiBRL at 5e7: **44%**)                         |
> | TrMRL [Melo, 2022]     | 14%              | 23% (5e7 steps) (both reported by [Shala, 2025])                 |
> | DME [Anonymous, 2025]  | 4%               | 26%
>
> Comparing with TrMRL [Melo, 2022] and ECET [Shala, 2025], which are based on computationally heavy transformer structures, all neural networks in GLiBRL are simple MLPs (we do not even have RNNs). GLiBRL achieves better results with efficient, simple neural architectures.
>
> To this end, we also address the concern about whether GLiBRL can be trained and tested on $\textbf{lower-end machines}$. The answer is $\textbf{Yes}$. Because of its lightweight neural architectures, GLiBRL is runnable on 8GB GPUs (tested on RTX 3070). The runtime for a single experiment (1 seed) will be less than 2 hours (see A.10 of the updated submission). Using higher-end GPUs / CPUs would further speed this up.
>
> $\textbf{(Q1)}$: *For how many seeds are you reporting the results?*
>
> $\textbf{Re (Q1)}$: As reported in the second paragraph of the experiments section, all of our experiments are run for 10 times using different seeds, following the setting in [McLean, 2025].
>
> $\textbf{(Q2)}$: *How large is the overhead of sampling for a Wishart distribution with GLiBRL?*
>
> $\textbf{Re (Q2)}$: The overhead of vanilla Wishart sampling is $O(D^3)$. When $D$ is relatively low, e.g., $D = D_T = 16$, sampling from Wishart is affordable. Otherwise, it can be slow. We have been working on ways to reduce the time complexity in the follow-up work.
>
> $\textbf{(Q3)}$: *How did you tune the hyperparameters of GLiBRL/How sensitive is GLiBRL with respect to hyperparameter choices?*
>
> $\textbf{Re (Q3)}$: As in $\textbf{Re (W1)}$, we have found GLiBRL not sensitive with respect to hyperparameter choices.
>
> $\textbf{(Q4)}$: *Why is Table 1 included? What does the max value tell us that is not shown from the final and average performances?*
>
> $\textbf{Re (Q4)}$: We thank you for your suggestion on the table. We initially had that just to report the exact numbers. In the updated version, we have changed the table to present comparisons between other recent methods, such as TrMRL, ECET and SDVT.
>
> We thank you again for being positive and encouraging on GLiBRL. GLiBRL has been further tested with your suggestions on recent work and hyperparameter sensitivity. We hope that our reply helps clarify the questions, and happy to provide further clarifications as needed.
>
> $\textbf{GLiBRL Authors}$

---

> > ### Author Response · Authors · 2025-11-19
> >
> > $\textbf{References:}$
> >
> > [Rakelly, 2019] Efficient Off-Policy Meta-Reinforcement Learning via Probabilistic Context Variables
> >
> > [McLean, 2025] Meta-World+: An Improved, Standardized, RL Benchmark
> >
> > [Zintgraf, 2021] VariBAD: Variational Bayes-Adaptive Deep RL via Meta-Learning
> >
> > [Lee, 2023] Parameterizing Non-Parametric Meta-Reinforcement Learning Tasks via Subtask Decomposition
> >
> > [Shala, 2025] Efficient Cross-Episode Meta-RL
> >
> > [Melo, 2022] Transformers are Meta-Reinforcement Learners
> >
> > [Anonymous, 2025] Dynamic Mixture Embeddings for Contextural Meta-Reinforcement Learning (https://openreview.net/pdf/d4fb544cd2d68e3a232bab49fdda1786c9e3f655.pdf)

---

> > ### Comment · Reviewer_qycn · 2025-11-20
> >
> > Thank you very much for your thorough responses to me and the other reviewers.
> > All my questions are clarified and I appreciate that GLiBRL can be run on lower-end machines. **I will increase my score** and would be happy to see the paper accepted.

---

> > > ### Author Response · Authors · 2025-11-20
> > >
> > > Thank you for the thoughtful assessment and for increasing the score. We appreciate your recognition and your constructive feedback.

---

### Official Review · Reviewer_Lz3W · 2025-10-31

**Soundness:** 3
**Presentation:** 3
**Contribution:** 3
**Rating:** 6
**Confidence:** 2

**Summary:**

The paper proposes GLiBRL, a Bayesian reinforcement learning framework that models dynamics and rewards as generalised linear functions of learnable basis features, with small neural networks producing features $C_T$ and $C_R$ and linear heads equipped with conjugate priors. This design yields closed form posteriors for model parameters and a tractable marginal log likelihood objective that replaces ELBO based training, aiming to avoid posterior collapse while preserving sample efficient adaptation.

**Strengths:**

1. The central idea of combining learnable feature maps with linear in features Bayesian heads is conceptually clean and technically sound, giving exact posteriors $p(\theta_T,\theta_R\mid \mathcal C)$ and a closed form objective $\log p_{\phi}(\mathcal C)$ that avoids the optimisation pathologies of ELBO methods. This brings welcome clarity to Bayesian RL by separating representation learning from conjugate inference.
2. The paper positions the approach against VariBAD and related meta RL models and provides evidence on standard ML10 and ML45 benchmarks that the method improves success rates and stabilises training.
3. The empirical section includes ablations that connect the full method to a Normal Normal special case and to variants without noise inference, helping isolate which components matter most, and the reported results are accompanied by enough hyperparameter detail to encourage reproducibility.

**Weaknesses:**

1. The linearity in learned feature space introduces an expressivity trade off; when true dynamics or rewards demand strongly nonlinear or interaction heavy structure, the burden shifts entirely to the feature networks. The main text does not deeply probe failure modes or provide capacity sweeps that link basis dimension to performance and stability.
2. The baseline set, while representative, omits some strong recent meta RL or Bayesian model based baselines and does not include PEARL like variants that more directly test the claim that conjugate training avoids ELBO issues in comparable settings.
3. Computational efficiency claims are qualitative. There is limited reporting of wall clock time, memory, and per episode update cost as a function of basis width and dataset size, and no scaling plots that demonstrate the practical advantage of closed form updates as feature dimension grows.
4. The probabilistic assumptions are restrictive, including Gaussian noise and independence between transition and reward parameters. The paper does not examine robustness to heteroscedastic or non Gaussian noise, nor does it discuss how the approach would adapt when the matrix normal row covariance is not identity, which could matter for real world dynamics.

**Questions:**

1. How sensitive are results to the capacities of the basis networks and to the feature dimensionalities $D_T$ and $D_R$; can you report systematic sweeps and identify thresholds where linear heads become the bottleneck?
2. Can you provide wall clock, memory, and per episode posterior update timings on ML10 and ML45, and a scaling study that varies basis width to validate that closed form updates remain advantageous as feature dimensionality grows?
3. How would the method handle non identity row covariances in a matrix normal likelihood, or heteroscedastic and heavy tailed noise; is there a reparameterisation or approximate conjugacy that preserves a tractable training objective, and what is the empirical impact on performance?
4. Since a key motivation is avoiding ELBO collapse, can you add diagnostics beyond expected KL, such as mutual information between latents and data or predictive log likelihood trajectories, and include a PEARL style comparator tuned on MetaWorld V3 to clarify whether GLiBRL’s advantage stems from conjugacy, feature learning, or both.

---

> ### Author Response · Authors · 2025-11-19
>
> Dear Reviewer Lz3W:
>
> Thank you for your feedback. We like to take this opportunity to address your concerns and answer your questions.
>
> $\textbf{(W1)}$: *The linearity in learned feature space introduces an expressivity trade off; ... The main text does not deeply probe failure modes or provide capacity sweeps that link basis dimension to performance and stability*.
>
> $\textbf{Re (W1)}$: We have included a sweep of task dimensions $D_T$ and $D_R$, see Appendix A.9 of the updated submission. We found that GLiBRL is stable and remains performant with related to changes in $D_T$ and $D_R$. We have also observed that with $D_T = 8$, GLiBRL has success rate of $\textbf{29}\%$, outperforming all the comparators.
>
> $\textbf{(W2)}$: *The baseline set, while representative, omits some strong recent meta RL or Bayesian model based baselines and does not include PEARL like variants that more directly test the claim that conjugate training avoids ELBO issues in comparable settings*.
>
> $\textbf{Re (W2)}$: We appreciate your suggestion on including more recent baselines. We have included comparisons with the following recent Meta-RL work in section 5.3 of the updated submission:
>
> | Method                 | ML10 (2e7 steps) | ML45 (9e7 steps)                                                 |
> |------------------------|------------------|------------------------------------------------------------------|
> | **GLiBRL**             | **25%**          | **45%**                                                          |
> | SDVT [Lee, 2023]       | 19%              | 20%                                                              |
> | ECET [Shala, 2025]     | 18%              | 38% (5e7 steps) (GLiBRL at 5e7: **44%**)                         |
> | TrMRL [Melo, 2022]     | 14%              | 23% (5e7 steps) (both reported by [Shala, 2025])                 |
> | DME [Anonymous, 2025]  | 4%               | 26%
>
> GLiBRL learns more efficiently and has success rates higher than that of all of these recent Meta-RL methods.
>
> It is also suggested that we should compare with PEARL / variants of PEARL as they will more directly test GLiBRL. First of all, our main comparator, VariBAD [Zintgraf, 2021], can be viewed as a PEARL-like variant as they are both deep Bayesian RL / task-inference based Meta-RL. In this sense, we have already compared with PEARL-like variant in the paper. Comparing with PEARL itself would also not be a more direct option, as VariBAD is more recent and achieves better results. Furthermore, we have already mentioned in the experiment section that PEARL performs poorly in the MetaWorld V3 [McLean, 2025] benchmark, with $< 3\%$ success rate in the ML10 benchmark with $\textbf{10}$ adaptation episodes and $\mathbf{5}\times$ more training steps than GLiBRL, which achieves $\sim25\%$ success rate $\textbf{zero-shot}$. The performance of GLiBRL can be better demonstrated comparing with other methods.
>
> $\textbf{(W3)}$:*Computational efficiency claims are qualitative. There is limited reporting of wall clock time, memory, and per episode update cost as a function of basis width and dataset size, and no scaling plots that demonstrate the practical advantage of closed form updates as feature dimension grows.*
>
> $\textbf{(Re W3)}$: GLiBRL is fast and lightweight. A single experiment on ML10 with $2e7$ training steps and $20$ evaluation steps takes less than $2$ hours with low-end GPUs such as RTX 3070, which also indicates that GLiBRL uses less than $8\text{GB}$ of GPU memory (see A.10 of the updated submission). As GLiBRL has already achieved SOTA performance with small task dimensions $D_T$ and $D_R$, as shown in Appendix A.9, we have found that increasing $D_T$ or $D_R$ only affects the training / testing time by a small amount.
>
> $\textbf{(W4)}$: *The probabilistic assumptions are restrictive, including Gaussian noise and independence between transition and reward parameters. The paper does not examine robustness to heteroscedastic or non-Gaussian noise, nor does it discuss how the approach would adapt when the matrix normal row covariance is not identity, which could matter for real world dynamics.*
>
> $\textbf{(Re W4)}$: Gaussian noise is a common assumption, appearing in most of the RL / Meta-RL / deep Bayesian RL work, due to its nice properties such as self-conjugacy. For the ease of modelling, many mainstream work (e.g., PEARL, VariBAD, SDVT [Lee, 2023]) even assume isotropic / diagonal covariance. In this sense, GLiBRL is more general as it does not have this assumption during the column covariance inference procedure. This allows GLiBRL to better capture the inter-dimensional relation of task variables $\theta_T, \theta_R$. Having this said, GLiBRL is open to non Gaussian noise as well, so long as conjugate prior functions are placed on $\theta_T, \theta_R$. GLiBRL has also been compared to models with heteroscedastic noise (i.e., VariBAD) in the submission, and substantial improvement has been shown.

---

> ### Author Response · Authors · 2025-11-19
>
> $\textbf{Re (W2.4) (cont'd)}$:
>
> Regarding the identity row covariance - it is also corresponding to a common assumption in the RL community, i.e. tuples of datapoints $(s, a, s', r)$ are $\textit{i.i.d.}$. Having this being said, GLiBRL can switch to arbitrary row covariance without any issues. We provide the non-identity row covariance case (of transitions, which can be adapted rewards easily) as follows, where $\mathbf{K}_T^{-1}$ is the arbitrary row covariance:
>
> $\textbf{Prior}$:
>
> $$p(\theta_T)  = \mathcal{MN} (T\_\mu | \mathbf{M}\_T, \boldsymbol{\Xi}\_T, T\_{\sigma}) \cdot \mathcal{W}(T\_{\sigma}^{-1}|\Omega\_{T}^{-1}, \nu\_{T})$$
>
> $\textbf{Likelihood}$:
>
> $$ p(\mathbf{S}\_i'|\mathbf{S}\_i, \mathbf{A}\_i, \theta\_T) = \mathcal{MN}(\mathbf{S}\_i' \vert {\mathbf{C}\_T T\_\mu}, {\mathbf{K}\_T^{-1}}, {T\_\sigma } ) $$
>
> $\textbf{Posterior}$:
>
> $$ p(\theta\_T|\mathbf{S}\_i, \mathbf{A}\_i, \mathbf{S}\_i') = \mathcal{MN}(T\_\mu|\mathbf{M}'\_T, \boldsymbol{\Xi}'\_T, T\_{\sigma})\cdot \mathcal{W}(T\_{\sigma}^{-1}|\Omega\_T'^{-1}, \nu'\_T)$$
>
> where
> $$ \mathbf{M}_T' = {\boldsymbol{\Xi}_T'}^{-1} \left[\mathbf{C}_T^\text{T} \mathbf{K}_T \mathbf{S}' + \boldsymbol{\Xi}_T \mathbf{M}_T \right]$$
> $${\boldsymbol{\Xi}_T'} =\mathbf{C}_T^\text{T}\mathbf{K}_T\mathbf{C}_T + \boldsymbol{\Xi}_T $$
> $${\boldsymbol{\Omega}_T'} = {\boldsymbol{\Omega}_T} + {\mathbf{S}'}^\text{T} \mathbf{K}_T \mathbf{S}'+ \mathbf{M}_T^\text{T}\boldsymbol{\Xi}_T \mathbf{M}_T -  {\mathbf{M}_T'}^\text{T} \boldsymbol{\Xi}_T' {\mathbf{M}_T'}$$
> $$\nu_T' = \nu_T + N $$
>
> The marginal log-likelihood can also be computed analytically. Hence, there would not be any issues considering non-identity row covariance.
>
> $\textbf{(Q1)}$: *How sensitive are results to the capacities of the basis networks $D_T$ and $D_R$ to the feature dimensionalities $D_T$ and $D_R$; can you report systematic sweeps and identify thresholds where linear heads become the bottleneck?*
>
> $\textbf{Re (Q1)}$: We have included Appendix A.9 that demonstrates GLiBRL is stable and remains performant with changes in $D_T$ and $D_R$.
>
> $\textbf{(Q2)}$: *Can you provide wall clock, memory, and per episode posterior update timings on ML10 and ML45, and a scaling study that varies basis width to validate that closed form updates remain advantageous as feature dimensionality grows?*
>
> $\textbf{Re (Q2)}$: As in $\textbf{(Re W3)}$, GLiBRL is both fast and lightweight. The time complexity of performing online inference is just quadratic, as shown in section 3.2 of the submission. Within lower task dimensions, the fast training / testing time does not vary by much.
>
> $\textbf{(Q3)}$: *How would the method handle non identity row covariances in a matrix normal likelihood, or heteroscedastic and heavy tailed noise; is there a reparameterisation or approximate conjugacy that preserves a tractable training objective, and what is the empirical impact on performance?*
>
> $\textbf{Re (Q3)}$: As discussed in $\textbf{(Re W4)}$, non-identity row covariance can be handled without any issues. The conjugacy is preserved, and the marginal likelihood can be optimised directly. GLiBRL has also been compared with VariBAD in the experiment section, whose noise model is heteroscedastic, and has shown improved performance. As we follow the common i.i.d. assumption of RL, which entails identity row covariances, we find it not necessary to run experiments using non-identity row covariance.
>
> $\textbf{(Q4)}$: *Since a key motivation is avoiding ELBO collapse, can you add diagnostics beyond expected KL, such as mutual information ... and include a PEARL style comparator tuned on MetaWorld V3 ....*
>
> $\textbf{Re (Q4)}$: We conclude on the discussion in $\textbf{Re (W2)}$: comparison to PEARL-like / PEARL has already been shown in the submission, and GLiBRL shows superior performance. The suggested mutual information metric is exactly defined as the expected KL divergence between the posterior and the prior, which means we have already done the comparison in the submission, as shown in the Appendix A.7.
>
> We hope our replies help in answering the questions and clarifying the capability and contribution of GLiBRL, and happy to provide further clarifications as needed.
>
> $\textbf{GLiBRL Authors}$
>
> $\textbf{References:}$
>
> [Rakelly, 2019] Efficient Off-Policy Meta-Reinforcement Learning via Probabilistic Context Variables
>
> [McLean, 2025] Meta-World+: An Improved, Standardized, RL Benchmark
>
> [Zintgraf, 2021] VariBAD: Variational Bayes-Adaptive Deep RL via Meta-Learning
>
> [Lee, 2023] Parameterizing Non-Parametric Meta-Reinforcement Learning Tasks via Subtask Decomposition
>
> [Shala, 2025] Efficient Cross-Episode Meta-RL
>
> [Melo, 2022] Transformers are Meta-Reinforcement Learners
>
> [Anonymous, 2025] Dynamic Mixture Embeddings for Contextural Meta-Reinforcement Learning (https://openreview.net/pdf/d4fb544cd2d68e3a232bab49fdda1786c9e3f655.pdf)

---

> ### Author Response · Authors · 2025-11-27
>
> Dear reviewer Lz3W:
>
> We are really grateful to have your insightful review. As the discussion phase is concluding in less than a week, we would like to check if our response has clarified the question, to make room for future discussions.
>
> Thanks,
>
> GLiBRL authors

---

### Official Review · Reviewer_Fbfd · 2025-11-01

**Soundness:** 2
**Presentation:** 3
**Contribution:** 2
**Rating:** 4
**Confidence:** 3

**Summary:**

This paper proposes the use of a linear model (of learned basis functions) for producing a parametrized distribution describing a family of tasks, for use in a Bayesian RL framework. The linearized final layer model enables efficient inference, and results in better performance than an existing method (VariBAD) and equivalent or slightly better performance than a meta-learning approach (MAML).

**Strengths:**

- Learning a distribution over a family of tasks that varies in a relatively low-dimensional way is a powerful approach to transfer, and has been only lightly explored. This is a reasonably practical approach to the problem.
 - The domains on which the method was applied are challening.
 - The approach seems mathematically solid, though I did not go through the math in detail.

**Weaknesses:**

First, the authors have not done a proper literature review. Their Background seems to be indicate that the field jumped straight from generic classical Bayesian RL (which the title implies this paper is, but which it is not, and which is correctly attributed to Duff) to meta-learning. In fact the setting the paper is in is either a latent-context MDP (due to Brunskill and Li, 2013 or so) or a hidden-parameter MDP (due to Doshi-Velez, 2016 or so), depending on whether the latent parameter generating the family of MDPs is discrete or continuous. The authors being unaware of these lines of work lead them to miss important related literature, for example the Bayesian hidden-parameter MDP model due to Killian (2017), which is probably the state-of-the-art model in this context, and very close to the model proposed here. An experimental comparison to that work, as well as follow-on work by Yao and Killian, is mandatory. For example, the authors mention sampling using the learned models as future work, but Killian already did that, eight years ago.

The experiments are also somewhat limited, being five samples over two domains, one of which does not show much improvement over an (at this point) very well established but quite generic meta-learning method.

**Questions:**

In your background section, do I understand correctly that by "context" you mean a batch of sample transitions from a single task? That is not what it means in a contextual MDP, or a latent context MDP, so some clarity here would be helpful. Perhaps the samples are a sort of non-parametric representation of the latent context.

---

> ### Author Response · Authors · 2025-11-19
>
> Dear Reviewer Fbfd:
>
> Thank you for the feedback. We would like to take this opportunity to address your concerns and questions.
>
> First, we would like to emphasise that the main contribution of GLiBRL is on the efficient and stable Bayesian inference side, and $\textit{not}$ on proposing new policy learners. Due to the efficient design of GLiBRL, both model-based (including more expensive POMDP solvers) and model-free methods can be used as policy learners of GLiBRL. In this paper, we use a model-free method, specifically PPO, as the policy learner. Compare to HiP-MDP [Doshi-Velez, 2016], including the Hip-MDP formulation of [Killian, 2017], GliBRL with PPO explicitly accounts for both unknown transition and unknown rewards, while Hip-MDP only accounts for unknown transition. Since the problem scenarios we test in this paper have unknown transition and reward functions, they cannot be run under the HiP-MDP framework and compared with GliBRL + PPO, fairly.  Therefore, we believe it would be better to compare with related work on recent deep BRL methods like PEARL [Rakelly, 2019] or VariBAD [Zintgraf, 2021], which enables reward learning.
>
> The following are clarifications to some of the concerns.
>
> $\textbf{(Q1)}$: *In your background section, do I understand correctly that by "context" you mean a batch of sample transitions from a single task?* ...
>
> $\textbf{Re (Q1)}$: Following the convention ([Rakelly, 2019], [Zintgraf, 2021]) in deep Bayesian RL, here "context" means tuples consisting of $(s, a, s', r)$. In the updated version of the paper, we have explicitly mentioned what contexts are ($\sim\text{line 114}$) to clarify the confusion.
>
> $\textbf{(W1)}$: *First, the authors have not done a proper literature review. Their Background seems to be indicate that the field jumped straight from generic classical Bayesian RL (which the title implies this paper is, but which it is not, and which is correctly attributed to Duff) to meta-learning*
>
> $\textbf{Re (W1)}$: We did not include HiP-MDPs initially due to space constraints. With page limits extended to 10 pages, we have added them in both background & related work sections. We renamed our title to Generalised Linear Models in $\textbf{Deep}$ Bayesian RL with Learnable Basis Functions, to clarify that we are addressing the inference in deep Bayesian RL problems.
>
> $\textbf{(W2.1)}$: *In fact the setting the paper is in is either a latent-context MDP (due to Brunskill and Li, 2013 or so) or a hidden-parameter MDP (due to Doshi-Velez, 2016 or so), .... The authors being unaware of these lines of work lead them to miss important related literature* ...
>
> $\textbf{Re (W2.1)}$: GLiBRL cannot be viewed as a HiP-MDP [Doshi-Velez, 2016], because the original work, as well as [Killian, 2017], [Yao, 2018] assume the reward function is fully known, while we do not have this assumption in GLiBRL. Furthermore, the Bayesian inference of GLiBRL is placed on context task variables $\theta_T, \theta_R$ directly, rather than on the weights of neural networks (detailed in $\textbf{Re (W2.3)}$).
>
> $\textbf{(W2.2)}$: ... *for example the Bayesian hidden-parameter MDP model due to Killian (2017), which is probably the state-of-the-art model in this context ...*
>
> $\textbf{Re (W2.2)}$: HiP-MDP framework, including the formulation in [Killian, 2017] and [Yao, 2018] assumes the reward function is known, while our paper is designed for problems where both transition and reward functions are unknown, thereby making a fair comparison with [Killian, 2017] and [Yao, 2018] impossible.
>
> Methods for Bayesian RL where both transition and reward functions are unknown have been proposed, with SOTA being PEARL [Rakelly, 2019] and VariBAD [Zintgraf, 2021], as per the comparison in recent Meta-RL papers, such as [Lee, 2023], [Melo, 2022], and [Shala, 2025]. In section 5.3 of the updated submission, we have added comparisons to these papers. (detailed in $\textbf{Re (W3)}$)

---

> ### Author Response · Authors · 2025-11-19
>
> $\textbf{(W2.3)}$ *[Killian, 2017] is very close to the model proposed here.*
>
> $\textbf{Re (W2.3)}$: We respectfully disagree that  [Killian, 2017] and GLiBRL are very close to each other. We list the differences below:
> - The main contribution of [Killian, 2017] is using Bayesian Neural Networks (BNNs) to relate latent contexts ($w_b$) to data, while the main contribution of GLiBRL is ELBO-free deep Bayesian RL. Neither [Killian, 2017] nor [Yao, 2018] can get away from approximate inference during training, while GLiBRL performs fully tractable exact inference.
> - In both [Killian, 2017] and [Yao, 2018], the Bayesian inference is placed on the weights of BNNs. This is significantly different from what we did: GLiBRL place Bayesian inference on the latent contexts ($\theta_T$, $\theta_R$) directly. In this sense, GLiBRL is closer to work like PEARL and VariBAD, rather than [Killian, 2017] or [Yao, 2018].
> - In both [Killian, 2017] and [Yao, 2018], the latent contexts $w_b$ are optimised with an MLE objective, instead of using Bayesian methods. The goal used in both work is the alpha-divergence between the transition posterior and prior, which in turn becomes an approximate Bayesian inference on the BNN weights. But note that the inference is placed on weights, not the latent contexts. Minimising the goal w.r.t. latent contexts $w_b$ directly (which is the approach both [Killian, 2017] and [Yao, 2018] took) is equivalently an MLE of $w_b$. When fixing the BNN weights during test time [Yao, 2018], the method losses most of the Bayesian features, hence cannot benefit from Bayes-optimal policies that provides principled ways of balancing exploration and exploitation [Ghavamzadeh, 2015]. Similar arguments are also made in the related work section of VariBAD [Zintgraf, 2021]. On the contrary, the Bayesian inference on latent contexts ($\theta_T$, $\theta_R$) in GLiBRL enables efficient test-time Bayes-optimal policies.
> - Finally, as mentioned in $\textbf{Re (W2.2)}$, [Killian, 2017] and [Yao, 2018] do not account for unknown reward while GliBRL does.
>
> Based on the above differences, we cannot say [Killian, 2017] is close to GLiBRL.
>
> $\textbf{(W2.4)}$: *An experimental comparison to that work, as well as follow-on work by Yao and Killian, is mandatory.*
>
> $\textbf{Re (W2.4)}$: Given our arguments in $\textbf{(W2.2)}$ and $\textbf{(W2.3)}$, we believe it is unfair to compare GliBRL with [Killian, 2017] or [Yao, 2018] because the assumption about what is unknown are different. There are recent and more related methods, such as PEARL and VariBAD, which we have compared GLiBRL with.
> Moreover, without running the experiments, practically, it is clear that [Killian, 2017] and [Yao, 2018] have a much higher computation requirement than GliBRL:
>
> - The use of BNNs is costly. As shown in [Killian, 2017] in page 12, "Training the BNN over the course of 300 separate episodes in the 2D toy domain was completed in a little more than 8 hours. " With each episode having 100 steps (page 4), this means $3e4$ steps have already taken 8 hours, even in a toy 2D domain. The standard training procedure of using Metaworld [McLean, 2025], which is currently one of the most challenging benchmark of Bayesian RL / Meta-RL, requires $\sim2e7$ steps, indicating prohibitively long training time if using [Killian, 2017]. With GLiBRL and all other comparators, $2e7$ steps can be finished in $1 \sim 2$ hours.
> - The weights of BNNs need to be updated during test time to achieve Bayes-optimal behaviour, as mentioned in $\textbf{Re (W2.3(3))}$. Again, the update of weights is much slower than that of Meta-RL methods such as MAML [Finn, 2017], as empirically shown in the test episode time of section D.2 of [Yang, 2020], where [Killian, 2017] is $\sim\mathbf{3000}\times$ slower than MAML in a toy example. In contrast, GLiBRL has test-time speed close to that of MAML. Such test-time efficiency also allows GLiBRL to handle dynamic environments much easier and faster.
>
> $\textbf{(W2.5)}$ *For example, the authors mention sampling using the learned models as future work, but Killian already did that, eight years ago.*
>
> $\textbf{Re (W2.5)}$: It is correct that [Killian, 2017] used a model-based policy learner. However, as mentioned in $\textbf{Re (W2.3(1))}$ and the abstract / introduction section of the submission, the contribution of GLiBRL is more on the tractable Bayesian inference side, and $\textbf{not}$ on the policy learning side. The policy learner in [Killian, 2017] is applicable in GLiBRL, but we will aim at proposing novel model-based methods that fully utilise the features of GLiBRL and hence are not applicable in previous methods due to their inaccurate (e.g., VariBAD) / slow (e.g., [Killian, 2017]) posterior update.

---

> ### Author Response · Authors · 2025-11-19
>
> $\textbf{(W3)}$ *The experiments are also somewhat limited, being five samples over two domains, one of which does not show much improvement over an (at this point) very well established but quite generic meta-learning method.*
>
> $\textbf{Re (W3)}$: We had 10 experiments with different seeds per method per benchmark, as mentioned in the second paragraph of the experiment section. It is worth mentioning that, five testing tasks is already extremely hard, as the agent has not seen those tasks at all during training, and the agent does not know which testing task it will be in during testing. The SOTA testing performance now in ML10 ($2e7$ steps) and ML45 ($9e7$ steps) is $29\\%$ (GLiBRL with $D_T = 8$) and $45\\%$ (GLiBRL), respectively.
>
> Meta-RL methods, such as RL2 [Duan, 2016] and MAML [Finn, 2017], achieve performance close to GLiBRL. However, we emphasize that these methods are not tracking latent context variables at all, hence does not have models to be used with sample-efficient model-based methods, nor as mentioned in the related work secion, uncertainty quantifications that can be very useful in fields like control or planning under uncertainty. Comparing against Bayesian RL methods that place (approximate) Bayesian inference on latent context variables, such as PEARL and VariBAD, our improvement is substantial by up to $2.7\times$ in success rate.
>
> To further demonstrate the strong performance of GLiBRL, we list success rates of more recent Meta-RL comparators below (also in the updated version of the submission):
>
> | Method                 | ML10 (2e7 steps) | ML45 (9e7 steps)                                                 |
> |------------------------|------------------|------------------------------------------------------------------|
> | **GLiBRL**             | **25%**          | **45%**                                                          |
> | SDVT [Lee, 2023]       | 19%              | 20%                                                              |
> | ECET [Shala, 2025]     | 18%              | 38% (5e7 steps) (GLiBRL at 5e7: **44%**)                         |
> | TrMRL [Melo, 2022]     | 14%              | 23% (5e7 steps) (both reported by [Shala, 2025])                 |
> | DME [Anonymous, 2025]  | 4%               | 26%                                                              |
>
> GLiBRL learns more efficiently and has success rates higher than that of all of these recent Meta-RL methods. With all these methods being compared, we do think GLiBRL has the SOTA performance.
>
> Last but not least, to the reviewer, we thank you again for providing thoughtful feedback that strengthened our work and reading through our replies. We hope that they help clarify the contribution and scope of GLiBRL.
>
> $\textbf{GLiBRL Authors}$
>
> $\textbf{References:}$
>
> [Killian, 2017] Robust and Efficient Transfer Learning with Hidden-Parameter Markov Decision Processes
>
> [Yao, 2018] Direct policy transfer via hidden parameter markov decision processes
>
> [Rakelly, 2019] Efficient Off-Policy Meta-Reinforcement Learning via Probabilistic Context Variables
>
> [McLean, 2025] Meta-World+: An Improved, Standardized, RL Benchmark
>
> [Zintgraf, 2021] VariBAD: Variational Bayes-Adaptive Deep RL via Meta-Learning
>
> [Doshi-Velez, 2016] Hidden Parameter Markov Decision Processes: A Semiparametric Regression Approach for Discovering Latent Task Parametrizations
>
> [Duan, 2016] Rl2: Fast reinforcement learning via slow reinforcement learning
>
> [Finn, 2017] Model-Agnostic Meta-Learning for Fast Adaptation of Deep Networks
>
> [Lee, 2021] Improving Generalization in Meta-RL with Imaginary Tasks from Latent Dynamics Mixture
>
> [Lee, 2023] Parameterizing Non-Parametric Meta-Reinforcement Learning Tasks via Subtask Decomposition
>
> [Yang, 2020] Single Episode Policy Transfer in Reinforcement Learning
>
> [Ghavamzadeh, 2015] Bayesian Reinforcement Learning: A Survey
>
> [Shala, 2025] Efficient Cross-Episode Meta-RL
>
> [Bing, 2024] Context-Based Meta-Reinforcement Learning With Bayesian Nonparametric Models
>
> [Beck, 2023] A Survey of Meta-Reinforcement Learning
>
> [Melo, 2022] Transformers are Meta-Reinforcement Learners
>
> [Anonymous, 2025] Dynamic Mixture Embeddings for Contextural Meta-Reinforcement Learning
> (https://openreview.net/pdf/d4fb544cd2d68e3a232bab49fdda1786c9e3f655.pdf)

---

> ### Author Response · Authors · 2025-11-27
>
> Dear reviewer Fbfd:
>
> We really appreciate your thoughtful review. As the discussion phase is concluding in less than a week, we would like to check if our response has clarified the question, to make room for future discussions.
>
> Thanks,
>
> GLiBRL authors

---

### Meta-Review · Area_Chair_u3Tn · 2026-01-06

**Summary:**

The reviewers raised concerns primarily around three areas: incomplete literature review regarding HiP-MDPs and related Bayesian RL work [Reviewer Fbfd], limited baseline comparisons with recent meta-RL methods [Reviewers qycn, Lz3W], and insufficient hyperparameter sensitivity analysis [Reviewers qycn, Lz3W]. The authors provided additional experiments comparing against TrMRL, ECET, SDVT, and DME, and added hyperparameter sweeps in Appendix A.9, which may satisfy Reviewer qycn.

**Reviewer Concerns:**

**Addressed by rebuttal:**
- Hyperparameter sensitivity concerns were addressed through new sweeps in Appendix A.9 showing GLiBRL remains stable across different settings for DT and DR [Reviewers qycn, Lz3W]
- Computational efficiency questions were partially addressed with runtime estimates (<2 hours on RTX 3070, <8GB memory) [Reviewer qycn]
- Additional baseline comparisons with recent methods (TrMRL, ECET, SDVT, DME) were added to Section 5.3 [Reviewers qycn, Lz3W]

**Outstanding concerns:**
- The claim that GLiBRL achieves "state-of-the-art" performance may be overstated given the mixed results across benchmarks [Reviewer qycn]
- [Minor] The literature review criticism regarding HiP-MDPs remains contentious; authors argued comparison is unfair due to different assumptions (unknown rewards vs. known rewards) [Reviewer Fbfd]

**Reviewer Scores:**

Reviewer Fbfd: 4. The reviewer may not be fully satisfied as their primary concern about mandatory comparisons with Killian (2017) and Yao (2018) was countered by authors citing different problem assumptions (unknown vs. known rewards).

Reviewer Lz3W: 6. The reviewer's questions about hyperparameter sweeps, computational costs, and PEARL-like comparisons were addressed. However the concerns about expressivity trade-offs and non-Gaussian noise handling may not be sufficiently addressed.

Reviewer qycn: 8. The reviewer explicitly stated "All my questions are clarified" and increased their score, appreciating that GLiBRL can run on lower-end machines and the thorough responses provided [Reviewer qycn].

---

### Decision · Program_Chairs · 2026-01-26

Reject